# ROSARL: REWARD-ONLY SAFE REINFORCEMENT LEARNING

## ABSTRACT

An important problem in reinforcement learning is designing agents that learn to solve tasks safely in an environment. A common solution is to define either a penalty in the reward function or a cost to be minimised when reaching unsafe states. However, designing reward or cost functions is non-trivial and can increase with the complexity of the problem. To address this, we investigate the concept of a *Minmax penalty*, the smallest penalty for unsafe states that leads to safe optimal policies, regardless of task rewards. We derive an upper and lower bound on this penalty by considering both environment *diameter* and *solvability*. Additionally, we propose a simple algorithm for agents to estimate this penalty while learning task policies. Our experiments demonstrate the effectiveness of this approach in enabling agents to learn safe policies in high-dimensional continuous control environments.

## 1 INTRODUCTION

Reinforcement learning (RL) has recently achieved success across a variety of domains, such as video games (Shao et al., 2019), robotics (Kalashnikov et al., 2018; Kahn et al., 2018) and autonomous driving (Kiran et al., 2021). However, if we hope to deploy RL in the real world, agents must be capable of completing tasks while avoiding unsafe or costly behaviour. For example, a navigating robot must avoid colliding with objects and actors around it, while simultaneously learning to solve the required task. Figure 1 shows an example.

Many approaches in RL deal with this problem by allocating arbitrary penalties to unsafe states when hand-crafting the reward function. However, the problem of specifying a reward function for desirable, safe behaviour is notoriously difficult (Amodei et al., 2016). *Importantly, penalties that are too small may result in unsafe behaviour, while penalties that are too large may result in increased learning times.* Furthermore, these rewards must be specified by an expert for each new task an agent faces. If our aim is to design truly autonomous, general agents, it is then simply impractical to require that a human designer specify penalties to guarantee optimal but safe behaviours for every task.

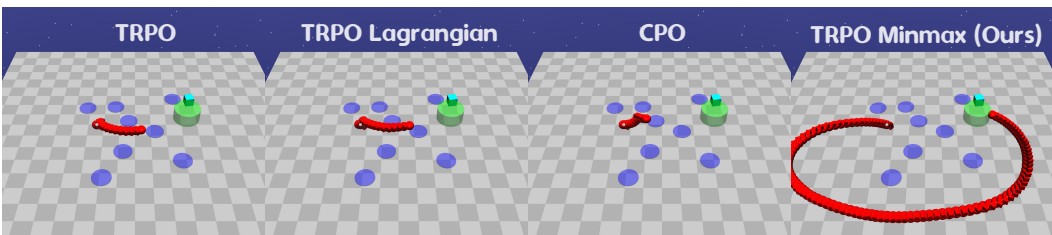

Figure 1: Sample trajectories of representative prior works—TRPO (Schulman et al., 2015) (leftmost), TRPO-Lagrangian (Ray et al., 2019) (middle-left), CPO (Achiam et al., 2017) (middle-right)—compared to ours (right-most) in the Safety Gym domain (Ray et al., 2019). For each, a point mass agent learns to reach a goal location (green cylinder) while avoiding unsafe regions (blue circles). The cyan block is a randomly placed movable obstacle. Our approach learns safer policies than the baselines, and works by simply changing the rewards received for entering unsafe regions to a learned penalty (keeping the rewards received for all other transitions unchanged).

When safety is an explicit goal, a common approach is to constrain policy learning according to some threshold on cumulative cost (Schulman et al., 2015; Ray et al., 2019; Achiam et al., 2017). While effective, these approaches require the design of a cost function whose specification can be as challenging as designing a reward function. Additionally, these methods may still result in unacceptably frequent constraint violations in practice, due to the large cost threshold typically used.

Rather than attempting to both maximise a reward function and minimise a cost function, which requires specifying both rewards and costs and a new learning objective, we should simply aim to have a better reward function—since we then do not have to specify yet another scalar signal nor change the learning objective. This approach is consistent with the *reward hypothesis* (Sutton & Barto, 2018) which states: " *All of what we mean by goals and purposes can be well thought of as maximisation of the expected value of the cumulative sum of a received scalar signal (reward).* " Therefore, the question we examine in this work is how to determine the *Minmax penalty*—the smallest penalty assigned to unsafe states such that the probability of reaching safe goals is maximised by an optimal policy. Rather than requiring an expert's input, we show that this penalty can be bounded by taking into account the *diameter* and *solvability* of an environment, and a practical estimate of it can be learned by an agent using its current value estimates. We make the following contributions:

(i) **Bounding the Minmax penalty**: We provide analytical upper and lower bounds on the Minmax penalty for unsafe transitions, and prove that using the upper bound results in policies that minimise the probability of reaching unsafe transitions (Theorem 2).

(ii) **Learning safety bounds**: We show that these bounds can be accurately estimated using policy evaluation (Sutton & Barto, 2018) (Theorem 1). Additionally, we show that estimating the Minmax penalty or bounds is NP-hard since it requires solving a longest path problem (Theorem 3).

(iii) **Learning safe policies**: Building on our theoretical analysis, we present a practical, model-free algorithm that allows agents to learn a sufficient penalty for unsafe transitions while simultaneously learning task policies (Algorithm 1). Since this approach only modifies the reward received for unsafe transitions, it is easily integrated into any existing RL pipeline that uses value-based methods.

## 2 BACKGROUND

We consider the typical RL setting where the task faced by an agent is modelled by a Markov Decision Process (MDP). An MDP is defined as a tuple $\langle \mathcal{S}, \mathcal{A}, P, R \rangle$, where $\mathcal{S}$ is a finite set of states, $\mathcal{A}$ is a finite set of actions, $P : \mathcal{S} \times \mathcal{A} \times \mathcal{S} \to [0\ 1]$ is the transition probability function, and $R : \mathcal{S} \times \mathcal{A} \times \mathcal{S} \to [R_{\text{MIN}}\ R_{\text{MAX}}]$ is the reward function. Our focus is on undiscounted MDPs that model stochastic shortest path problems (Bertsekas & Tsitsiklis, 1991) in which an agent must reach some goals in the non-empty set of absorbing states $\mathcal{G} \subset \mathcal{S}$. The set of non-absorbing states $\mathcal{S} \setminus \mathcal{G}$ are referred to as *internal states*. We will also refer to the tuple $\langle \mathcal{S}, \mathcal{A}, P \rangle$ as the environment, and the MDP $\langle \mathcal{S}, \mathcal{A}, P, R \rangle$ as a task to be solved. The agent is then associated with a *policy* $\pi : \mathcal{S} \to \mathcal{A}$ which it uses to take actions in the environment. The quality of a policy is usually defined by its *value function* $V^\pi(s) = \mathbb{E}^\pi[\sum_{t=0}^\infty R(s_t, a_t, s_{t+1})]$, which specifies the expected return under that policy starting from state $s$.

**Standard RL:** The standard goal of an agent is to learn an optimal policy $\pi^*$ that maximises the value function $V^{\pi^*}(s) = \max_\pi V^\pi(s)$ for all $s \in \mathcal{S}$. Since tasks are undiscounted, $\pi^*$ is guaranteed to exist by assuming that the value function of *improper policies* is unbounded from below—where *proper policies* are those that are guaranteed to reach an absorbing state (Van Niekerk et al., 2019). Since there always exists a deterministic $\pi^*$ (Sutton & Barto, 1998), and $\pi^*$ is proper, we will focus our attention on the set of all deterministic proper policies $\Pi$.

**Safe RL:** This setting is typically modelled in prior works by a constrained Markov Decision Process (CMDP) $\langle \mathcal{S}, \mathcal{A}, P, R, K, l \rangle$, which augments an MDP with a cost function $K : \mathcal{S} \times \mathcal{A} \times \mathcal{S} \to \mathbb{R}$ and a cost threshold $l \in \mathbb{R}$ (Altman, 1999). Here, a given policy $\pi$ can also be characterised by its cost value function $V_K^\pi(s) = \mathbb{E}^\pi[\sum_{t=0}^\infty K(s_t, a_t, s_{t+1})]$, and the policy is *feasible* if $V_K^\pi(s) \leq l$ for all $s \in \mathcal{S}$. Where $\widehat{\Pi}$ is the set of all feasible policies, the goal of an agent here is now to learn an optimal safe policy $\widehat{\pi}^*$ that maximises the value function $V^{\widehat{\pi}^*}(s) = \max_{\widehat{\pi} \in \widehat{\Pi}} V^{\widehat{\pi}}(s)$ for all $s \in \mathcal{S}$ (Ray et al., 2019). To ensure that $\widehat{\pi}^*$ exists and is well defined, $\widehat{\Pi}$ must not be empty, which means that $K$ and $l$ must be chosen carefully such that there exists a policy $\pi$ that satisfies $V_K^\pi(s) \leq l$ for all $s \in \mathcal{S}$.

**ROSARL (Ours):** In contrast to most prior works, in this work we are interested in learning safe policies without the need to specify cost functions and cost thresholds. In particular, we are interested in learning policies that can maximise rewards while avoiding unsafe transitions, where any unsafe transition immediately leads to termination in a set of unsafe absorbing states $\mathcal{G}^! \subset \mathcal{G}$. Since some environments may have no policy that avoids unsafe transitions with probability 1, we formally define a safe policy as a proper policy that minimises the probability of unsafe transitions (Definition 1). Hence, where $\widehat{\Pi}$ is the set of all safe policies, the goal of an agent in this work is to learn an optimal safe policy $\widehat{\pi}^*$ that maximises the value function $V^{\widehat{\pi}^*}(s) = \max_{\widehat{\pi} \in \widehat{\Pi}} V^{\widehat{\pi}}(s)$ for all $s \in \mathcal{S}$.

**Definition 1** *Consider an environment $\langle \mathcal{S}, \mathcal{A}, P \rangle$ with unsafe states $\mathcal{G}^! \subset \mathcal{G}$. Where $s_T$ is the final state of a trajectory starting from state $s$, let $P_s^\pi(s_T \in \mathcal{G}^!)$ be the probability of reaching $\mathcal{G}^!$ from $s$ under a proper policy $\pi \in \Pi$. Then $\pi$ is called safe if $\pi \in \arg\min_{\pi' \in \Pi} P_s^{\pi'}(s_T \in \mathcal{G}^!)$ for all $s \in \mathcal{S}$.*

# 3 AVOIDING UNSAFE ABSORBING STATES

Given an environment, we aim to bound the smallest penalty (hence the largest reward) to use for unsafe transitions to guarantee optimal safe policies. We define this penalty as the Minmax penalty $R_{\text{Minmax}}$, which is the largest reward for unsafe transitions that lead to optimal safe policies:

**Definition 2** *Consider an environment $\langle \mathcal{S}, \mathcal{A}, P \rangle$ where task rewards $R(s, a, s')$ are bounded by $[R_{MIN} \ R_{MAX}]$ for all $s' \notin \mathcal{G}^!$. Let $\pi^*$ be an optimal policy for one such task $\langle \mathcal{S}, \mathcal{A}, P, R \rangle$. We define the Minmax penalty of this environment as the scalar $R_{Minmax} \in \mathbb{R}$ that satisfies the following:*

*(i) If $R(s, a, s') < R_{Minmax}$ for all $s' \in \mathcal{G}^!$, then $\pi^*$ is safe for all $R$;*

*(ii) If $R(s, a, s') > R_{Minmax}$ for some $s' \in \mathcal{G}^!$ reachable from $\mathcal{S} \setminus \mathcal{G}$, then there exists an $R$ s.t. $\pi^*$ is unsafe.*

Hence, the Minmax penalty represents the boundary where on one side *no* reward function has an optimal policy that is unsafe, and on the other *there exist* a reward function with an optimal policy that is unsafe. Interestingly, when $R(s, a, s') = R_{\text{Minmax}}$, there may exist optimal safe and unsafe policies simultaneously—hence no RL algorithm with such rewards can be guaranteed to converge to optimal safe policies. We next demonstrate this using the Chain-walk running example.

## 3.1 A MOTIVATING EXAMPLE: THE CHAIN-WALK ENVIRONMENT

To illustrate the difficulty in designing reward functions for safe behaviour, consider the simple *chain-walk* environment in Figure 2a. It consists of four states $s_0, s_1, s_2, s_3$ where $\mathcal{G} = \{s_1, s_3\}$ and $\mathcal{G}^! = \{s_1\}$. The agent has two actions $a_1, a_2$, the initial state is $s_0$, and the diagram denotes the transition probabilities. Task rewards for safe transitions are bounded by $[R_{\text{MIN}} \ R_{\text{MAX}}] = [-1 \ 0]$. The absorbing transitions have a reward of 0 while all other transitions have a reward of $R_{step} = -1$, and the agent must reach the goal state $s_3$, but not the unsafe state $s_1$. Hence, the question here is what penalty to give for transitions from $s_0$ into $s_1$ such that the optimal policies are safe. Figures 2b-2d exemplify how too large penalties result in longer convergence times, while too small ones result in unsafe policies, demonstrating the need to find the Minmax penalty.

Since the transitions per action can be stochastic, controlled by $p_1, p_2 \in [0 \ 1]$, and $s_3$ is further from the start state $s_0$ than $s_1$, the agent may not always be able to avoid $s_1$. Consider for example the deterministic case when $p_1 = p_2 = 0$. For any penalty less than $-2$ for transitions into $s_1$, the optimal policy in $s_0$ is to always pick $a_2$ which always reaches $s_1$. For a sufficiently high penalty for reaching $s_1$ (any penalty higher than $-2$), the optimal policy in $s_0$ is to always pick action $a_1$, which always reaches $s_3$. Interestingly, if the penalty is exactly $-2$, then both action $a_1$ (safe transition to $s_2$) and action $a_2$ (unsafe transition to $s_1$) are optimal—hence an RL algorithm here will not necessarily converge to the optimal safe action $a_1$. Additionally, for $p_1 = p_2 = 0.4$ (Figure 2c), a higher penalty is required for $a_1$ to stay optimal in state $s_0$.

To capture this relationship between the stochasticity of an environment and the required penalty to obtain safe policies, we introduce a notion of *solvability*, which measures the ability of an agent to reach safe goals. Additionally, observe that as $p_2$ increases, the probability that the agent can

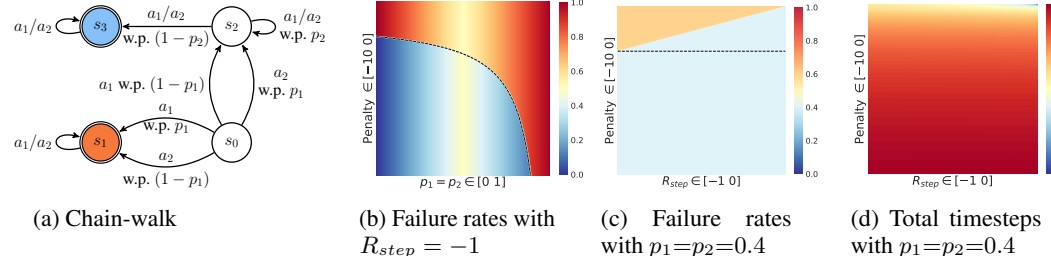

(a) Chain-walk

(b) Failure rates with $R_{step} = -1$

(c) Failure rates with $p_1 = p_2 = 0.4$

(d) Total timesteps with $p_1 = p_2 = 0.4$

Figure 2: The effect of different choices of penalty for unsafe transitions ($s_0$ to $s_1$) on optimal policies in the chain-walk environment. (a) The transition probabilities of the chain-walk environment (where $p_1, p_2 \in [0\ 1]$); (b) The failure rate for each penalty in $[-10\ 0]$ and each transition probabilities ($p_1 = p_2 \in [0\ 1]$), with a task reward of $R_{step} = -1$; (c) The failure rate for each penalty in $[-10\ 0]$ and each task reward in $[-1\ 0]$, with transition probabilities given by $p_1 = p_2 = 0.4$; (d) The total timesteps needed to learn optimal policies to convergence (using value iteration (Sutton & Barto, 1998)) for each penalty in $[-10\ 0]$ and each task reward in $[-1\ 0]$, with transition probabilities given by $p_1 = p_2 = 0.4$. The black dashed lines in (b) and (c) show the Minmax penalty.

transition from $s_2$ to $s_3$ decreases—thereby increasing the number of timesteps spent to reach the goal. Therefore, the penalty for $s_1$ must also consider the environment's *diameter* to ensure an optimal policy will not simply reach $s_1$ to avoid self-transitions in $s_2$.

## 3.2 On the Diameter and Solvability of Environments

Clearly, the size of the penalty that needs to be given for unsafe states depends on the *size* of the environment. We define this size as the *diameter* of the environment, which is the highest expected timesteps to reach an absorbing state from an internal state when following a proper policy:

**Definition 3** *Define the diameter of an environment as* $D := \max\limits_{s \in \mathcal{S} \setminus \mathcal{G}} \max\limits_{\pi \in \Pi} \mathbb{E}\left[T(s_T \in \mathcal{G} | \pi)\right],$ *where* $T(s_T \in \mathcal{G} | \pi)$ *is the timesteps taken to reach* $\mathcal{G}$ *from* $s$ *when following a proper policy* $\pi$.

This definition of diameter is similar to the one used in Auer et al. (2008), except that here we are maximising over deterministic proper policies instead of minimising over all deterministic policies. Given this diameter, a possible natural choice for the reward for unsafe states is to give a penalty that is as large as receiving the smallest task reward for the longest path to safe goal states: $\bar{R}_{\text{MAX}} := R_{\text{MIN}} D'$, where $D'$ is the diameter for safe policies $D' := \max\limits_{s \in \mathcal{S} \setminus \mathcal{G}} \max\limits_{\pi \in \Pi} \mathbb{E}\left[T(s_T \in \mathcal{G} \setminus \mathcal{G}^! | \pi)\right]$. However, while $\bar{R}_{\text{MAX}}$ aims to make reaching unsafe states worse than reaching safe goals, it does not consider the solvability of an environment, nor the possibility that an unsafe policy receives $R_{\text{MAX}}$ everywhere in its trajectory. We can formally define the solvability of an environment as follows:

**Definition 4** *Define the degree of solvability as* $C := \min\limits_{s \in S \setminus \mathcal{G}} \min\limits_{\substack{\pi \in \Pi \\ P_s^\pi(s_T \notin \mathcal{G}^!) \neq 0}} P_s^\pi(s_T \notin \mathcal{G}^!).$

$C$ measures the degree of solvability of the environment by simply taking the smallest non-zero probability of reaching safe goal states by following a proper policy. For example, if the dynamics are deterministic, then any deterministic policy $\pi$ will either reach a safe goal or not. That is, $P_s^\pi(s_T \notin \mathcal{G}^!)$ will either be 0 or 1. Since we require $P_s^\pi(s_T \notin \mathcal{G}^!) \neq 0$, it must be that $C = 1$. Consider, for example, the chain-walk environment with different choices for $p$. Since actions in $s_2$ do not affect the transition probability, there are only 2 relevant deterministic policies $\pi_1(s) \mapsto a_1$ and $\pi_2(s) \mapsto a_2$. This gives $P_{s_1}^{\pi_1}(s_T \notin \mathcal{G}^!) = (1 - p_1)\mathbb{1}(p_2 = 1)$ and $P_{s_1}^{\pi_2}(s_T \notin \mathcal{G}^!) = p_1\mathbb{1}(p_2 = 1)$. Here, $C = 1$ when $p_1 = p_2 = 0$ because the task is deterministic and $s_3$ is reachable. $C$ then tends to 0.5 as $p_1$ and $p_2$ gets closer to 0.5, making the environment uniformly random. Finally, the environment is not solvable when $p = 1$ since $s_3$ is unreachable from $s_2$. Hence we can also think of $C = 0$ as the *limit* of $C$ when safe goals are unreachable. Interestingly, this means that in deterministic environments our definition of solvability is similar to *reachability* in temporal-logic tasks—where there may or may not exist a policy that satisfies a task specification (Tasse et al., 2022).

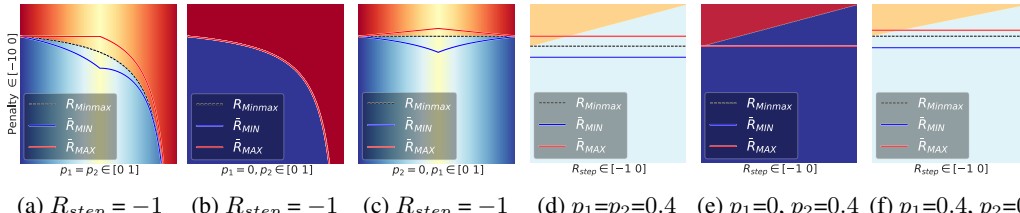

(a) $R_{step} = -1$    (b) $R_{step} = -1$    (c) $R_{step} = -1$    (d) $p_1 = p_2 = 0.4$    (e) $p_1 = 0$, $p_2 = 0.4$    (f) $p_1 = 0.4$, $p_2 = 0$

Figure 3: Failure rates of optimal policies in the chain-walk environment. We show the effect of stochasticity ($p_1$ and $p_2$) and task rewards ($R_{step}$) on the bounds ($\bar{R}_{\text{MIN}}$ and $\bar{R}_{\text{MAX}}$) of the Minmax penalty ($R_{\text{Minmax}}$). The solvability and diameter for the bounds are estimated using Algorithm 2.

Given the diameter and solvability of an environment, we can now define a choice for the Minmax penalty that takes into account both $D$, $C$, and $R_{\text{MAX}}$: $\bar{R}_{\text{MIN}} := (R_{\text{MIN}} - R_{\text{MAX}})\frac{D}{C}$. This choice of penalty says that since stochastic shortest path tasks require an agent to learn to achieve desired terminal states, if the agent enters an unsafe terminal state, it should receive the largest penalty possible by a proper policy. We now investigate the effect of these penalties on the failure rate of optimal policies.

### 3.3 ON THE FAILURE RATE OF OPTIMAL POLICIES

We begin by proposing a simple model-based algorithm for estimating the diameter and solvability, from which the penalties are then obtained. We describe the method here and present the pseudo-code in **Algorithm 2** in Appendix B. Here, the diameter is estimated as follows: (i) For each deterministic policy $\pi$, estimate its expected timesteps $T(s_T \in \mathcal{G})$ (or $T(s_T \in \mathcal{G} \setminus \mathcal{G}^!)$ for $D'$) by using policy evaluation (Sutton & Barto, 2018) with rewards of 1 at all internal states; (ii) Then, calculate $D$ using the equation in Definition 3. Similarly, the solvability is estimated by estimating the reach probability $P_s^\pi(s_T \notin \mathcal{G}^!)$ of each deterministic policy $\pi$ using rewards of 1 for transitions into safe goal states and zero otherwise. This approach converges via the convergence of policy evaluation (**Theorem 1**).

**Theorem 1 (Estimation)** *Algorithm 2 converges to $D$ and $C$ for any given solvable environment.*

Figure 3 shows the result of applying this algorithm in the chain-walk MDP. Here, $R_{\text{Minmax}}$ is compared to accounting for $D$ only ($\bar{R}_{\text{MAX}}$) and accounting for both $C$ and $D$ ($\bar{R}_{\text{MIN}}$). Interestingly, we can observe $\bar{R}_{\text{MIN}} \leq R_{\text{Minmax}}$ and $\bar{R}_{\text{MAX}} \geq R_{\text{Minmax}}$ consistently, highlighting how considering the diameter only is insufficient to guarantee optimal safe policies. It also indicates that these penalties may bound $R_{\text{Minmax}}$ in general. We show in **Theorem 2** that this is indeed the case.

**Theorem 2 (Safety Bounds)** *Consider a solvable environment where task rewards are bounded by $[R_{MIN}\ R_{MAX}]$ for all $s' \notin \mathcal{G}^!$. Then $\bar{R}_{MIN} \leq R_{Minmax} \leq \bar{R}_{MAX}$.*

Theorem 2 says that for any MDP whose rewards for unsafe transitions are bounded above by $\bar{R}_{\text{MIN}}$, the optimal policy both minimises the probability of reaching unsafe states and maximises the probability of reaching safe goal states. Hence, any penalty $\bar{R}_{\text{MIN}} - \epsilon$, where $\epsilon > 0$ can be arbitrarily small, will guarantee optimal safe policies. Similarly, the theorem shows that any reward higher than $\bar{R}_{\text{MAX}}$ may have optimal policies that do not minimise the probability of reaching unsafe states. These can be observed in Figure 3. The figure demonstrates why considering both the diameter and solvability of an MDP is necessary to guarantee safe policies, because the diameter alone does not always minimise the failure rate.

## 4 PRACTICAL ALGORITHM FOR LEARNING SAFE POLICIES

While the Minmax penalty of an MDP can be accurately estimated using policy evaluation (Algorithm 2), it requires knowledge of the environment dynamics (or an estimate of it). These are difficult quantities to estimate from an agent's experience, which is further complicated by the need to also learn the true optimal policy for the estimated Minmax penalty. Hence, obtaining an accurate estimate of the Minmax penalty is impractical in model-free and function approximation settings where the state and action spaces are large. In fact, it is NP-hard since it depends on the diameter, which requires solving a longest-path problem.

**Theorem 3 (Complexity)** *Estimating the Minmax penalty $R_{Minmax}$ accurately is NP-hard.*

Given the above challenges, we require a practical method for learning the Minmax penalty. Ideally, this method should require no knowledge of the environment dynamics and should easily integrate with existing RL approaches. To achieve this, we first note that $(R_{\text{MIN}} - R_{\text{MAX}})\frac{D}{C} = (DR_{\text{MIN}} - DR_{\text{MAX}})\frac{1}{C} = (V_{\text{MIN}} - V_{\text{MAX}})\frac{1}{C}$, where $V_{\text{MIN}}$ and $V_{\text{MAX}}$ are the value function bounds. Hence, a practical estimate of the Minmax penalty can be efficiently learned by estimating the value gap $V_{\text{MIN}} - V_{\text{MAX}}$ using observations of the reward and the agent's estimate of the value function. **Algorithm 1** shows the full pseudo-code. The agent here receives a reward $r_t$ after each environment interaction and updates its estimate of the reward bounds $R_{\text{MIN}} \leftarrow \min(R_{\text{MIN}}, r_t)$ and $R_{\text{MAX}} \leftarrow \max(R_{\text{MAX}}, r_t)$, the value bounds $V_{\text{MIN}} \leftarrow \min(V_{\text{MIN}}, R_{\text{MIN}}, V(s_t))$ and $V_{\text{MAX}} \leftarrow \max(V_{\text{MAX}}, R_{\text{MAX}}, V(s_t))$, and the Minmax penalty $\bar{R}_{\text{MIN}} \leftarrow V_{\text{MIN}} - V_{\text{MAX}}$, where $V(s_t)$ is the learned value function at time step $t$. We note how the solvability $C$ is also not explicitly considered in this estimate of $\bar{R}_{\text{MIN}}$, since it is also expensive to estimate. Instead, given that the main purpose of $C$ is to make $\bar{R}_{\text{MIN}}$ more negative the more stochastic the environment is, we notice that this is already achieved in practice by the reward and value estimates. Since $R_{\text{MIN}}$ is estimated using $R_{\text{MIN}} \leftarrow \min(R_{\text{MIN}}, r_t)$, then every time the agent enters an unsafe state, we have that: $r_t \leftarrow \bar{R}_{\text{MIN}}$, $R_{\text{MIN}} \leftarrow \bar{R}_{\text{MIN}}$, and then $\bar{R}_{\text{MIN}} \leftarrow \bar{R}_{\text{MIN}} - V_{\text{MAX}}$. This means that when the estimated $V_{\text{MAX}}$ is greater than zero, the penalty estimate $\bar{R}_{\text{MIN}}$ become more negative every time the agent enters an unsafe state.

Finally, whenever an agent encounters an unsafe state, the reward can be replaced by $\bar{R}_{\text{MIN}}$ to disincentivise unsafe behaviour. Since $V_{\text{MAX}}$ is estimated using $V_{\text{MAX}} \leftarrow \max(V_{\text{MAX}}, R_{\text{MAX}}, V(s_t))$, it leads to an optimistic estimation of $\bar{R}_{\text{MIN}}$. Hence, we observe no need to add an $\epsilon > 0$ to $\bar{R}_{\text{MIN}}$.

---

**Algorithm 1:** RL while learning Minmax penalty

---

**Input** : RL algorithm **A**, max timesteps $T$
**Initialise :** $R_{\text{MIN}} = 0, R_{\text{MAX}} = 0, V_{\text{MIN}} = R_{\text{MIN}}, V_{\text{MAX}} = R_{\text{MAX}}, \pi$ and $V$ as per **A**
   **for** t in T **do**
      **observe** a state $s_t$, **take** an action $a_t$ using $\pi$ as per **A**, and **observe** $s_{t+1}, r_t$
      $R_{\text{MIN}}, R_{\text{MAX}} \leftarrow \min(R_{\text{MIN}}, r_t), \max(R_{\text{MAX}}, r_t)$
      $V_{\text{MIN}}, V_{\text{MAX}} \leftarrow \min(V_{\text{MIN}}, R_{\text{MIN}}, V(s_t)), \max(V_{\text{MAX}}, R_{\text{MAX}}, V(s_t))$
      $\bar{R}_{\text{MIN}} \leftarrow V_{\text{MIN}} - V_{\text{MAX}}$
      $r_t \leftarrow \bar{R}_{\text{MIN}}$ **if** $s_{t+1} \in \mathcal{G}^!$ **else** $r_t$
      **update** $\pi$ and $V$ with $(s_t, a_t, s_{t+1}, r_t)$ as per **A**
   **end for**

---

## 5 EXPERIMENTS

While the theoretical Minmax penalty is guaranteed to lead to optimal safe policies, it is unclear whether this also holds for the practical estimate proposed in Section 4. Hence, this section aims to investigate three main natural questions regarding the proposed practical algorithm (see the Appendix for more domain details and experiments): (i) How does Algorithm 1 behave when the theoretical assumptions are satisfied? (ii) How does Algorithm 1 behave when the theoretical assumptions are *not* satisfied? (iii) How does Algorithm 1 compare to prior approaches towards Safe RL? For each result, we report the mean (solid line) and one standard deviation around it (shaded region).

### 5.1 BEHAVIOUR WHEN THEORY HOLDS

For this experiment, we consider the Russell & Norvig (2016) LAVA GRIDWORLD domain (Figure 4a). It satisfies the setting we assumed in Section 2 since it is a stochastic shortest path with finite states and actions. To test our approach, we modify Q-learning (Watkins, 1989) with $\epsilon$-greedy exploration such that the agent updates its estimate of the Minmax penalty as learning progresses and uses it as the reward whenever the lava state is reached, following the procedure outlined in Section 4. The action-value function is initialised to 0 for all states and actions, $\epsilon = 0.1$ and the learning rate $\alpha = 0.1$.

We examine the performance of our modified Q-learning approach across three values of the slip probability ($sp$) of the LAVA GRIDWORLD. A slip probability of 0 represents a fully deterministic

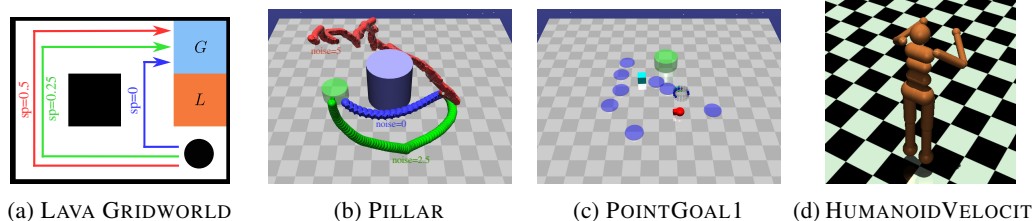

|  |  |  |  |
| :---: | :---: | :---: | :---: |
| (a) LAVA GRIDWORLD | (b) PILLAR | (c) POINTGOAL1 | (d) HUMANOIDVELOCITY |

Figure 4: Experimental domains. The agent here must: (a) Navigate (in $|\mathcal{S}| = 11$) to the goal $G$ while avoiding the lava $L$ . (b) Navigate (in $|\mathcal{S}| = \mathbb{R}^{60}$) to the green cylinder 🟢 while avoiding the pillar 🔵. See Appendix D for detailed descriptions. (c-d) These are standard Safety-Gymnasium tasks, only modified to terminate when an unsafe state is reached: entering a blue region in POINTGOAL1 ($|\mathcal{S}| = \mathbb{R}^{60}$) and exceeding the velocity threshold in HUMANOIDVELOCITY ($|\mathcal{S}| = \mathbb{R}^{348}$).

environment, while a slip probability of 0.5 represents a more stochastic environment. Results are plotted in Figure 5. In the case of the fully deterministic environment, the Minmax penalty bound obtained via Algorithm 2 is $\bar{R}_{\text{MIN}} = -9.9$, since $C = 1$ and $D = 9$ (see Figure 5a for the other values of $sp$). However, the agent is able to learn a relatively smaller penalty ($-1.1$ in Figure 5b) to consistently minimise failure rate and maximise returns (Figures 5c and 5d). The resulting optimal policy then chooses the shorter path that passes near the lava location ($sp = 0$ in Figure 4a). As the stochasticity of the environment increases (leading to significant increase in $R_{\text{MIN}}$), a larger penalty is learned to incentivise longer, safer policies. Interestingly, we also see that Q-learning using the oracle penalty $R_{\text{MIN}}$ still successfully minimising failure rates but struggles to maximise rewards (possibly requiring much longer training time to converge due to the large penalty). We can, therefore, conclude that while there is a gap between the true Minmax penalty and the one learned via Algorithm 1, this algorithm can still learn optimal safe policies when the theoretical setting holds.

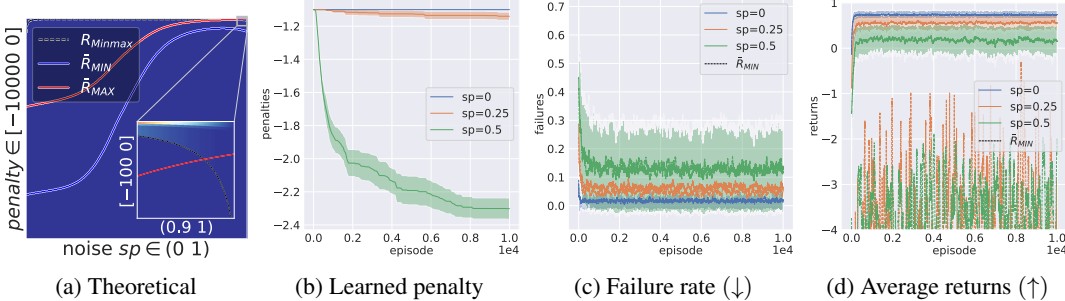

|  |  |  |  |
| :---: | :---: | :---: | :---: |
| (a) Theoretical | (b) Learned penalty | (c) Failure rate ($\downarrow$) | (d) Average returns ($\uparrow$) |

Figure 5: Effect of increase in the slip probability (sp) of the LAVA GRIDWORLD on the learned Minmax penalty and corresponding failure rate and returns. (a) shows the failure rates for varying fixed penalties, and the theoretical bounds. (b-c) Shows the performance of the agent when learning its own penalty and when using the theoretical bound $R_{\text{MIN}}$. The results are averaged over 20 seeds.

## 5.2 BEHAVIOUR WHEN THEORY DOES NOT HOLD

For this experiment, we consider the Safety Gym (Ray et al., 2019) PILLAR domain (Figure 4b). It does not satisfy the setting we assumed in Section 2 since it is continuous and not a shortest path task[1]. To test our approach in this setting, we modify TRPO (Schulman et al., 2015) (denoted TRPO-Minmax) to use the estimate of the Minmax penalty as described in Algorithm 1.

We examine the performance of TRPO-Minmax for five levels of $noise$ in the PILLAR environment, similarly to the experiments in Section 5.1. Results are plotted in Figure 6. We observe similar results to Section 5.1, where the agent uses its learned Minmax penalty (Figure 6a) to successfully

---

[1]The PILLAR domain does not satisfy the formal shortest shortest path setting we assume since: it is discounted and policies that do not reach $\mathcal{G}$ are not guaranteed to have value functions that are unbounded from below (due to the default dense rewards in Safety Gym which positively rewards moving towards the goal).

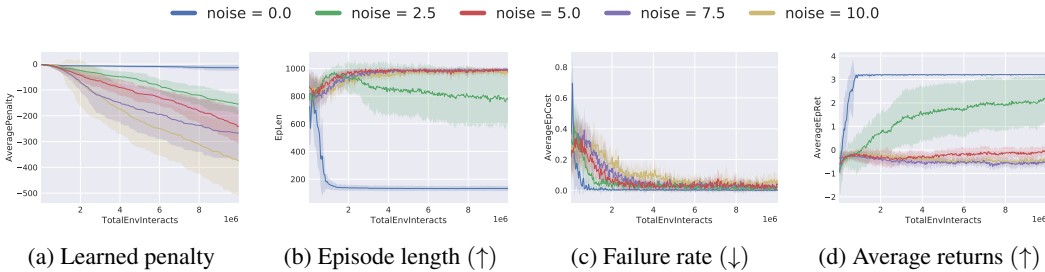

| (a) Learned penalty | (b) Episode length (↑) | (c) Failure rate (↓) | (d) Average returns (↑) |

Figure 6: Performance of TRPO-Minmax in the PILLAR environment with varying noise levels. Each training run is over 10 million steps and the results are averaged over 10 seeds.

learn safe policies (Figure 6c) while solving the task (Figure 6d), using safer longer paths for more noisy dynamics (Figures 4b,6b). Interestingly, it also correctly prioritises low failure rates when the dynamics are too noisy to safely reach the goal ($noise \geq 5$). This is in contrast to the behaviour of prior works which learn risky policies, prioritising the maximisation of rewards over short trajectories that are highly likely to result in collisions (as shown in Figures 8-9 and Table 10). We can, therefore, conclude that Algorithm 1 can learn safe policies even in discounted high-dimensional continuous-control domains requiring function approximation.

### 5.3 COMPARISON TO REPRESENTATIVE BASELINES

For this experiment, we consider representative OmniSafe baselines (Ji et al., 2024) in various Safety-gymnasium domains (Ji et al., 2023), including the PILLAR domain. As a baseline representative of typical RL approaches, we use Trust Region Policy Optimisation (TRPO) (Schulman et al., 2015). To represent constraint-based approaches and PID Lagrangian methods, we compare against, TRPO with Lagrangian constraints (TRPO-Lagrangian) (Ray et al., 2019), Sauté RL with TRPO (Sauté-TRPO) (Sootla et al., 2022), Constrained Proximal Policy Optimisation with PID (CPPOPID) (Stooke et al., 2020), and Penalised Proximal Policy Optimisation (P3O) Zhang et al. (2022). All baselines use the implementations provided by Ji et al. (2024), and form a set of widely used baselines in safety domains (Zhang et al., 2020; Sootla et al., 2022; Yang et al., 2023; Ji et al., 2024). As in Ji et al. (2024), all approaches use feed-forward MLPs, value networks of size (256,256), and $tanh$ activation functions. The cost threshold for the constrained algorithms is set to 0, the best we found. The experiments are run over 10 million episodes and averaged over 10 runs.

We compare the performance of TRPO-Minmax (Ours) to that of the baselines in Safety-Gymnasium domains. Tables 1-2 shows the results. We observe that in the deterministic case $noise = 0$ of the PILLAR task, all the algorithms achieve similar performance, successfully maximising returns while minimising the failure rates—except P3O which struggles relatively more to perfectly satisfy the constraints. However, for the stochastic cases $noise > 0$, all the baselines struggle significantly to minimise the probability of unsafe transitions as they prioritise maximising returns. In contrast, the results obtained show that TRPO-Minmax successfully solves the tasks while minimising failures for both deterministic and stochastic environments. In addition, we can observe from the success rates, returns and episode lengths in POINTGOAL1 that TRPO-Minmax chooses longer paths to the goal when that is required to maximise safety, even going against the dense environment rewards. This demonstrates its ability to trade off between rewards maximisation and safety, with a strong bias towards safety—in contrast to the baselines which seem strongly biased towards reward maximisation. This general behaviour is also consistent even when using other baseline implementations (Ray et al., 2019), where TRPO-Minmax interestingly takes longer to learn to maxise rewards but consistently prioritises safer trajectories (Figures 1,10-13).

## 6 RELATED WORK

Guiding agents toward desirable behaviors has been explored through reward shaping, which augments reward functions to improve learning efficiency but requires that the optimal policy is unaltered (Ng et al., 1999; Devidze et al., 2021). This is undesirable in safe RL where the optimal

| Method | Noise 0.0 | | Noise 1.5 | | Noise 2.5 | |
|---|---|---|---|---|---|---|
| | Failure ↓ | Returns ↑ | Failure ↓ | Returns ↑ | Failure ↓ | Returns ↑ |
| Ours | **0.00 ± 0.00** | **3.21 ± 0.00** | **0.06 ± 0.02** | **3.01 ± 0.08** | **0.14 ± 0.05** | **2.61 ± 0.27** |
| TRPO-L | **0.00 ± 0.00** | **3.21 ± 0.00** | 0.09 ± 0.04 | 2.90 ± 0.12 | 0.20 ± 0.05 | 2.38 ± 0.46 |
| S-TRPO | **0.00 ± 0.00** | 3.20 ± 0.03 | 0.11 ± 0.04 | 2.81 ± 0.14 | 0.19 ± 0.09 | 2.45 ± 0.54 |
| TRPO | **0.00 ± 0.00** | **3.21 ± 0.00** | 0.13 ± 0.08 | 2.74 ± 0.29 | 0.28 ± 0.10 | 2.05 ± 0.52 |
| CPPOPID | **0.00 ± 0.00** | **3.21 ± 0.00** | 0.17 ± 0.08 | 2.61 ± 0.31 | 0.28 ± 0.16 | 2.12 ± 0.64 |
| P3O | 0.10 ± 0.30 | 2.74 ± 1.01 | 0.11 ± 0.13 | 2.43 ± 1.04 | 0.17 ± 0.07 | 2.42 ± 0.34 |

Table 1: Comparison in the PILLAR environment with varying noise. We train using 10 random seeds for 10 million steps and evaluate each over 100 random seeds, for a total of 1000 evaluation episodes. Ours is TRPO-Minmax, while TRPO-L is TRPO-Lagrangian and S-TRPO is Sauté-TRPO.

| Method | POINTGOAL1 | | | | HUMANOIDVELOCITY | |
|---|---|---|---|---|---|---|
| | Failure ↓ | Success ↑ | Returns ↑ | Length ↑ | Failure ↓ | Returns ↑ |
| Ours | **0.08 ± 0.05** | **0.45 ± 0.16** | 1.87 ± 1.60 | **950 ± 31.45** | **0.01 ± 0.01** | 610 ± 54 |
| TRPO-L | 0.62 ± 0.07 | 0.34 ± 0.06 | 10.17 ± 0.77 | 607 ± 56.15 | 0.09 ± 0.04 | 800 ± 100 |
| S-TRPO | 0.79 ± 0.03 | 0.21 ± 0.03 | **11.01 ± 0.55** | 493 ± 24.51 | 0.35 ± 0.12 | **957 ± 165** |
| TRPO | 0.78 ± 0.05 | 0.22 ± 0.05 | 10.68 ± 0.74 | 483 ± 33.07 | 0.43 ± 0.18 | 944 ± 206 |
| P3O | 0.56 ± 0.07 | 0.42 ± 0.07 | 10.63 ± 0.74 | 667 ± 49.14 | 0.02 ± 0.06 | 611 ± 43 |

Table 2: Comparison in default Safety-Gymnasium tasks (with termination only at unsafe states). We train using 10 random seeds for 10 million steps and evaluate each over 100 random seeds, for a total of 1000 evaluation episodes. Ours is TRPO-Minmax, while TRPO-L is TRPO-Lagrangian and S-TRPO is Sauté-TRPO. See Appendix H for ablations of terminal states and training curves.

policy may be unsafe according to some safety constraints. More popularly, works in constrained RL usually impose safety constraints to limit cost violations while maximizing rewards (Ray et al., 2019; Sootla et al., 2022). In contrast, our work optimizes terminal state rewards to minimize undesirable behaviors directly. Finally, other works like Shielding complements these approaches by using model or human interventions to prevent unsafe actions (Dalal et al., 2018; Wagener et al., 2021; Tennenholtz et al., 2022). As shielding typically modifies transition dynamics rather than reward functions, it aligns naturally with our reward-focused framework. See Appendix C for an expended related works.

# 7 DISCUSSION AND FUTURE WORK

This paper investigates a new approach towards safe RL by asking the question: *Is a scalar reward enough to solve tasks safely?* To answer this question, we bound the Minmax penalty, which takes into account the diameter and solvability of an environment in order to minimise the probability of encountering unsafe states. We prove that the penalty does indeed minimise this probability, and present a method that uses an agent's value estimates to learn an estimate of the penalty. Our results in tabular and high-dimensional continuous settings have demonstrated that, by encoding the safe behaviour directly in the reward function via the Minmax penalty, agents are able to solve tasks while prioritising safety, learning safer policies than popular constraint-based approaches. Our method is also easy to incorporate with any off-the-shelf RL algorithms that maintain value estimates, requiring no changes to the algorithms themselves. By autonomously learning the penalty, our method also alleviates the need for a human designer to manually tweak rewards or cost functions to elicit safe behaviour.

Finally, while we show that scalar rewards are indeed enough for safe RL, the current analysis is only applicable to unsafe terminal states—which only covers tasks that can be naturally represented by stochastic-shortest path MDPs. Given that other popular RL settings like discounted MDPs can be converted to stochastic shortest path MDPs (Bertsekas, 1987; Sutton & Barto, 1998), a promising future direction could be to find the dual of our results for other theoretically equivalent settings. In conclusion, we see this reward-only approach as a promising direction towards truly autonomous agents capable of independently learning to solve tasks safely.

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

## A   PROOFS OF THEORETICAL RESULTS

**Theorem 1 (Estimation)** *Algorithm 2 converges to $D$ and $C$ for any given solvable environment.*

**Proof** This follows from the convergence guarantee of policy evaluation (Sutton & Barto, 1998). ∎

**Theorem 2 (Safety Bounds)** *Consider a solvable environment where task rewards are bounded by $[R_{MIN}\ R_{MAX}]$ for all $s' \notin \mathcal{G}^!$. Then $\bar{R}_{MIN} \le R_{Minmax} \le \bar{R}_{MAX}$.*

**Proof** Let $\pi^*$ be an optimal policy for an arbitrary task $\langle \mathcal{S}, \mathcal{A}, P, R \rangle$ in the environment. Given the definition of the Minmax penalty (Definition 2), we need to show the following:

  (i) If $R(s, a, s') < \bar{R}_{\text{MIN}}$ for all $s' \in \mathcal{G}^!$, then $\pi^*$ is safe for all $R$; and
  (ii) If $R(s, a, s') > \bar{R}_{\text{MAX}}$ for some $s' \in \mathcal{G}^!$ reachable from $\mathcal{S} \setminus \mathcal{G}$, then there exists an $R$ s.t. $\pi^*$ is unsafe.

(i) Since $\pi^*$ is optimal, it is also proper and hence must reach $\mathcal{G}$.

Assume $\pi^*$ is unsafe. Then there exists another proper policy $\pi$ that is safe, such that

$$P_s^\pi(s_T \in \mathcal{G}^!) < P_s^{\pi^*}(s_T \in \mathcal{G}^!) \quad \text{for some } s \in \mathcal{S}.$$

Then,

$$V^{\pi^*}(s) \ge V^\pi(s)$$

$$\implies \mathbb{E}_s^{\pi^*}\left[\sum_{t=0}^{\infty} R(s_t, a_t, s_{t+1})\right] \ge \mathbb{E}_s^{\pi}\left[\sum_{t=0}^{\infty} R(s_t, a_t, s_{t+1})\right]$$

$$\implies \mathbb{E}_s^{\pi^*}\left[G^{T-1} + R(s_T, a_T, s_{T+1})\right] \ge \mathbb{E}_s^{\pi}\left[G^{T-1} + R(s_T, a_T, s_{T+1})\right],$$

$$\text{where } G^{T-1} = \sum_{t=0}^{T-1} R(s_t, a_t, s_{t+1}) \text{ and } T \text{ is a random variable denoting when } s_{T+1} \in \mathcal{G}.$$

$$\implies \mathbb{E}_s^{\pi^*}\left[G^{T-1}\right] + \left(P_s^{\pi^*}(s_T \notin \mathcal{G}^!)R(s_T, a_T, s_{T+1}) + P_s^{\pi^*}(s_T \in \mathcal{G}^!)\bar{R}_{\text{unsafe}}(s_T, a_T, s_{T+1})\right)$$

$$\ge \mathbb{E}_s^{\pi}\left[G^{T-1}\right] + \left(P_s^{\pi}(s_T \notin \mathcal{G}^!)R(s_T, a_T, s_{T+1}) + P_s^{\pi}(s_T \in \mathcal{G}^!)\bar{R}_{\text{unsafe}}(s_T, a_T, s_{T+1})\right),$$

$$\text{where } \bar{R}_{\text{unsafe}} \text{ denotes the rewards for transitions into } \mathcal{G}^! \text{ and } a_T = \pi^*(s_T).$$

$$\implies \mathbb{E}_s^{\pi^*}\left[G^{T-1}\right] + \left(P_s^{\pi^*}(s_T \notin \mathcal{G}^!)R(s_T, a_T, s_{T+1}) + \bar{R}_{\text{unsafe}}(s_T, a_T, s_{T+1})\right)$$

$$\ge \mathbb{E}_s^{\pi}\left[G^{T-1}\right] + \left(P_s^{\pi}(s_T \notin \mathcal{G}^!)R(s_T, a_T, s_{T+1}) + P_s^{\pi}(s_T \in \mathcal{G}^!)\bar{R}_{\text{unsafe}}(s_T, a_T, s_{T+1})\right),$$

$$\implies \mathbb{E}_s^{\pi^*}\left[G^{T-1}\right] + \left(1 - P_s^{\pi}(s_T \in \mathcal{G}^!)\right)\bar{R}_{\text{unsafe}}(s_T, a_T, s_{T+1})$$

$$\ge \mathbb{E}_s^{\pi}\left[G^{T-1}\right] + \left(P_s^{\pi}(s_T \notin \mathcal{G}^!) - P_s^{\pi^*}(s_T \notin \mathcal{G}^!)\right)R(s_T, a_T, s_{T+1})$$

$$\implies \mathbb{E}_s^{\pi^*}\left[G^{T-1}\right] + \left(1 - P_s^{\pi}(s_T \in \mathcal{G}^!)\right)\bar{R}_{\text{MIN}}$$

$$> \mathbb{E}_s^{\pi}\left[G^{T-1}\right] + \left(P_s^{\pi}(s_T \notin \mathcal{G}^!) - P_s^{\pi^*}(s_T \notin \mathcal{G}^!)\right)R(s_T, a_T, s_{T+1}),$$

$$\text{since } \bar{R}_{\text{unsafe}}(s_T, a_T, s_{T+1}) < \bar{R}_{\text{MIN}}.$$

$$\implies \mathbb{E}_s^{\pi^*}\left[G^{T-1}\right] + \left(1 - P_s^{\pi}(s_T \in \mathcal{G}^!)\right)(R_{\text{MIN}} - R_{\text{MAX}})\frac{D}{C}$$

$$> \mathbb{E}_s^{\pi}\left[G^{T-1}\right] + \left(P_s^{\pi}(s_T \notin \mathcal{G}^!) - P_s^{\pi^*}(s_T \notin \mathcal{G}^!)\right)R(s_T, a_T, s_{T+1})$$

$$\implies \mathbb{E}_s^{\pi^*}\left[G^{T-1}\right] + (R_{\text{MIN}} - R_{\text{MAX}})D$$

$$> \mathbb{E}_s^{\pi}\left[G^{T-1}\right] + \left(P_s^{\pi}(s_T \notin \mathcal{G}^!) - P_s^{\pi^*}(s_T \notin \mathcal{G}^!)\right)R(s_T, a_T, s_{T+1}), \text{ using definition of } C.$$

$$\implies \mathbb{E}_s^{\pi^*}\left[G^{T-1}\right] - R_{\text{MAX}}D$$

$$> \mathbb{E}_s^{\pi} \left[ G^{T-1} \right] + \left( P_s^{\pi}(s_T \notin \mathcal{G}^!) - P_s^{\pi^*}(s_T \notin \mathcal{G}^!) \right) R(s_T, a_T, s_{T+1}) - R_{\text{MIN}}D$$

$$\implies \mathbb{E}_s^{\pi^*} \left[ G^{T-1} \right] - R_{\text{MAX}}D > 0,$$

$$\text{since } \mathbb{E}_s^{\pi} \left[ G^{T-1} \right] + \left( P_s^{\pi}(s_T \notin \mathcal{G}^!) - P_s^{\pi^*}(s_T \notin \mathcal{G}^!) \right) R(s_T, a_T, s_{T+1}) \geq R_{\text{MIN}}D$$

$$\implies \mathbb{E}_s^{\pi^*} \left[ G^{T-1} \right] > R_{\text{MAX}}D.$$

But this is a contradiction since the expected return of following an optimal policy up to a terminal state without the reward for entering the terminal state must be less than receiving $R_{\text{MAX}}$ for every step of the longest possible trajectory to $\mathcal{G}$. Hence we must have $\pi^* \in \arg\min_{\pi} P_s^{\pi}(s_T \in \mathcal{G}^!)$.

*(ii)* Assume $\pi^*$ is safe. Then, $P_s^{\pi^*}(s_T \notin \mathcal{G}^!) \geq P_s^{\pi'}(s_T \notin \mathcal{G}^!)$ for all $s \in \mathcal{S}$, $\pi' \in \Pi$.

Let $\pi$ be the policy that maximises the probability of reaching $s' \in \mathcal{G}^!$ from some state $s \in \mathcal{G}$. Then, similarly to (i), we have

$$V^{\pi^*}(s) \geq V^{\pi}(s)$$

$$\implies \mathbb{E}_s^{\pi^*} \left[ G^{T-1} \right] + \left( P_s^{\pi^*}(s_T \in \mathcal{G}^!) - P_s^{\pi}(s_T \in \mathcal{G}^!) \right) \bar{R}_{\text{unsafe}}(s_T, a_T, s_{T+1})$$

$$\geq \mathbb{E}_s^{\pi} \left[ G^{T-1} \right] + \left( P_s^{\pi}(s_T \notin \mathcal{G}^!) - P_s^{\pi^*}(s_T \notin \mathcal{G}^!) \right) R(s_T, a_T, s_{T+1})$$

$$\implies \mathbb{E}_s^{\pi} \left[ G^{T-1} \right] + \left( P_s^{\pi}(s_T \in \mathcal{G}^!) - P_s^{\pi^*}(s_T \in \mathcal{G}^!) \right) \bar{R}_{\text{unsafe}}(s_T, a_T, s_{T+1})$$

$$\leq \mathbb{E}_s^{\pi^*} \left[ G^{T-1} \right] + \left( P_s^{\pi^*}(s_T \notin \mathcal{G}^!) - P_s^{\pi}(s_T \notin \mathcal{G}^!) \right) R(s_T, a_T, s_{T+1})$$

$$\implies \mathbb{E}_s^{\pi} \left[ G^{T-1} \right] + \left( P_s^{\pi}(s_T \in \mathcal{G}^!) - P_s^{\pi^*}(s_T \in \mathcal{G}^!) \right) \bar{R}_{\text{MAX}}$$

$$< \mathbb{E}_s^{\pi^*} \left[ G^{T-1} \right] + \left( P_s^{\pi^*}(s_T \notin \mathcal{G}^!) - P_s^{\pi}(s_T \notin \mathcal{G}^!) \right) R(s_T, a_T, s_{T+1}), \text{ since } \bar{R}_{\text{unsafe}} > \bar{R}_{\text{MAX}}.$$

$$\implies \mathbb{E}_s^{\pi} \left[ G^{T-1} \right] + \left( P_s^{\pi}(s_T \in \mathcal{G}^!) - P_s^{\pi^*}(s_T \in \mathcal{G}^!) \right) R_{\text{MIN}}D'$$

$$< \mathbb{E}_s^{\pi^*} \left[ G^{T-1} \right] + \left( P_s^{\pi^*}(s_T \notin \mathcal{G}^!) - P_s^{\pi}(s_T \notin \mathcal{G}^!) \right) R(s_T, a_T, s_{T+1}), \text{ by definition of } \bar{R}_{\text{MAX}}.$$

$$\implies \mathbb{E}_s^{\pi} \left[ G^{T-1} \right] + R_{\text{MIN}}D'$$

$$< \mathbb{E}_s^{\pi^*} \left[ G^{T-1} \right] + \left( P_s^{\pi^*}(s_T \notin \mathcal{G}^!) - P_s^{\pi}(s_T \notin \mathcal{G}^!) \right) R(s_T, a_T, s_{T+1})$$

$$\implies \mathbb{E}_s^{\pi} \left[ G^{T-1} \right] + R_{\text{MIN}}D' < 0$$

But this is a contradiction when $R$ is such that the agent receives a reward of $R_{\text{MAX}} \geq |R_{\text{MIN}}|D'$ at least once in its trajectory when following $\pi$ and zero everywhere else.

∎

**Theorem 3 (Complexity)** *Estimating the Minmax penalty $R_{Minmax}$ accurately is NP-hard.*

**Proof** This follows from the NP-hardness of longest-path problems. Since the Minmax penalty is bounded by $\bar{R}_{\text{MIN}}$ and $\bar{R}_{\text{MAX}}$, both are defined by the diameter, which is in turn defined as the expected total timesteps of the longest path. ∎

# B ALGORITHMS

---

**Algorithm 2:** Estimating the Diameter and Solvability

---

**Input** : $\langle \mathcal{S}, \mathcal{A}, P \rangle$, $R_D(s') := \mathbb{1}(s' \notin \mathcal{G})$, $R_C(s, a, s') := \mathbb{1}(s \notin \mathcal{G} \text{ and } s' \in \mathcal{G} \setminus \mathcal{G}^!)$

**Initialise** : Diameter $D = 0$, Solvability $C = 1$, Value functions $V_D^\pi(s) = 0$, $V_C^\pi(s) = 0$, Error $\Delta = 1$

```
for π ∈ Π do
    /* Policy evaluation for D */
    while Δ > 0 do
        Δ ← 0
        for s ∈ S do
            v' ← ∑_{s'} P(s'|s, π(s))(R_D(s')+V_D^π(s'))

            Δ = max{Δ, |V_D^π(s) − v'|}
            V_D^π(s) ← v'
        end for
    end while
    for s ∈ S do
        D = max{D, V_D^π(s)}
    end for
end for
```

```
for π ∈ Π do
    /* Policy evaluation for C */
    while Δ > 0 do
        Δ ← 0
        for s ∈ S do
            v' ← ∑_{s'} P(s'|s, π(s))(R_C(s, π(s), s')+V_C^π(s'))

            Δ = max{Δ, |V_C^π(s) − v'|}
            V_C^π(s) ← v'
        end for
    end while
    for s ∈ S do
        C = min{C, V_C^π(s)} if V_C^π(s) ≠ 0 else
        C
    end for
end for
```

---

# C EXTENDED RELATED WORK

## C.1 REWARD SHAPING

The problem of designing reward functions to produce desired policies in RL settings is well-studied (Singh et al., 2009). Particular focus has been placed on the practice of *reward shaping*, in which an initial reward function provided by an MDP is augmented in order to improve the rate at which an agent learns the same optimal policy (Ng et al., 1999; Devidze et al., 2021). While sacrificing some optimality, other approaches like Lipton et al. (2016) propose shaping rewards using an idea of intrinsic fear. Here, the agent trains a supervised fear model representing the probability of reaching unsafe states in a fixed horizon, scales said probabilities by a fear factor, and then subtracts the scaled probabilities from Q-learning targets.

These approaches differ from ours in that they seek to find reward functions that improve convergence while preserving the optimality from an initial reward function. In contrast, we seek to determine the optimal rewards for terminal states in order to minimise undesirable behaviours irrespective of the original reward function and optimal policy.

## C.2 CONSTRAINED RL

Disincentivising or preventing undesirable behaviours is core to the field of safe RL. A popular approach is to define constraints on the behaviour of an agent using CMDPs, tasking the agent with limiting the accumulation of costs associated with violating safety constraints while simultaneously maximising reward (Altman, 1999; Achiam et al., 2017; Chow et al., 2018; Ray et al., 2019; Hasan-zadeZonuzy et al., 2021). Widely used examples of these approaches include constrained policy optimisation (CPO) (Achiam et al., 2017), which augments TRPO (Schulman et al., 2015) with constraints to satisfy a constrained MDP, and TRPO-Lagrangian (Ray et al., 2019), which combines Lagrangian methods with TRPO. Another example is Sauté RL (Sootla et al., 2022), which incorporates the cost function into the rewards and augments the state with the remaining "cost budget" spent by violating safety constraints. Other constraint-based approaches include Projection-based CPO (Yang et al., 2020), which projects a TRPO policy onto a space defined by constraints, and PID Lagrangian methods (Stooke et al., 2020), which augment Lagrangian methods with PID control.

In deterministic environments with a cost threshold of 0, the set of safe policies for these approaches are the same as ours. However, in stochastic environments, these approaches require the correct choice of inequality constraints to even be well defined. If the cost threshold is not carefully chosen, there may exist no policy that satisfies the CMDP constraints, implying there would exist no optimal safe policy to converge to. For example, in the LAVA GRIDWORLD or the PILLAR domains with $noise > 0$, a cost threshold of 0 can never be satisfied by any policy for all states, making these approaches theoretically ill-defined in these environments with that cost threshold. That said, we found in practice that a cost threshold of 0 gave them the best performance in the safety-gym experiments (compared to 1 and the default of 25). In contrast, we showed the existence of a Minmax penalty irrespective of the stochasticity of the environment. Additionally, while these approaches in general theoretically define or learn safety parameters—like Lagrange coefficients—for each reward function even when the cost function and cost threshold remain unchanged, our minmax penalty approach is theoretically defined and learned for all reward functions.

## C.3 SHIELDING

Finally, another important line of work involves relying on interventions from a model (Dalal et al., 2018; Wagener et al., 2021) or human (Tennenholtz et al., 2022) to prevent unsafe actions from being considered by the agent (shielding the agent) or prevent the environment from executing those unsafe actions by correcting them (shielding the environment). Other approaches here also look at using temporal logics to define or enforce safety constraints on the actions considered or selected by the agent (Alshiekh et al., 2018).

These approaches fit seamlessly into our proposed reward-only framework since they are primarily about modifications on the transition dynamics and not the reward function—for example, unsafe actions here can simply lead to unsafe goal states.

# D  EXPERIMENTS DETAILS

**LAVA GRIDWORLD**   This is a gridworld with 11 positions ($|\mathcal{S}| = 11$) and 4 cardinal actions ($|\mathcal{A}| = 4$). The agent here must reach a goal location $G$ while avoiding a lava location $L$ (hence $\mathcal{G} = \{L, G\}$ and $\mathcal{G}^! = \{L\}$). A wall is also present in the environment and, while not unsafe, must be navigated around. The environment has a *slip probability (sp)*, so that with probability $sp$ the agent's action is overridden with a random action. The agent receives $R_{\text{MAX}} = +1$ reward for reaching the goal, as well as $R_{step} = -0.1$ reward at each timestep to incentivise taking the shortest path to the goal.

**Safety Gym PILLAR**   This is a custom Safety Gym domain in which the simple point robot must navigate to a goal location 🟢 around a large pillar 🔵 (hence $\mathcal{G} = \{$🟢,🔵$\}$ and $\mathcal{G}^! = \{$🔵$\}$). All details of the environment are the same as in Ray et al. (2019) except when stated otherwise. Just as in Ray et al. (2019), the agent uses *pseudo-lidar* to observe the distance to objects around it ($|\mathcal{S}| = \mathbb{R}^{60}$), and the action space is continuous over two actuators controlling the direction and forward velocity ($|\mathcal{A}| = [-1, 1]^2$). This direction and forward velocity can be noisy, determined by a $noise$ scalar as follows: $a_{new} = a + (noise)a_{noise}$ where $a_{new}$ is the new direction and forward velocity, $a \in \mathcal{A}$ is the agent's action, and $a_{noise} \in \mathcal{A}$ is a uniformly sampled random vector. The goal, pillar, and agent locations remain unchanged for all episodes. Each episode terminates once the agent reaches the goal or collides with the pillar (with a reward of $-1$). Otherwise, episodes terminate after 1000 timesteps.

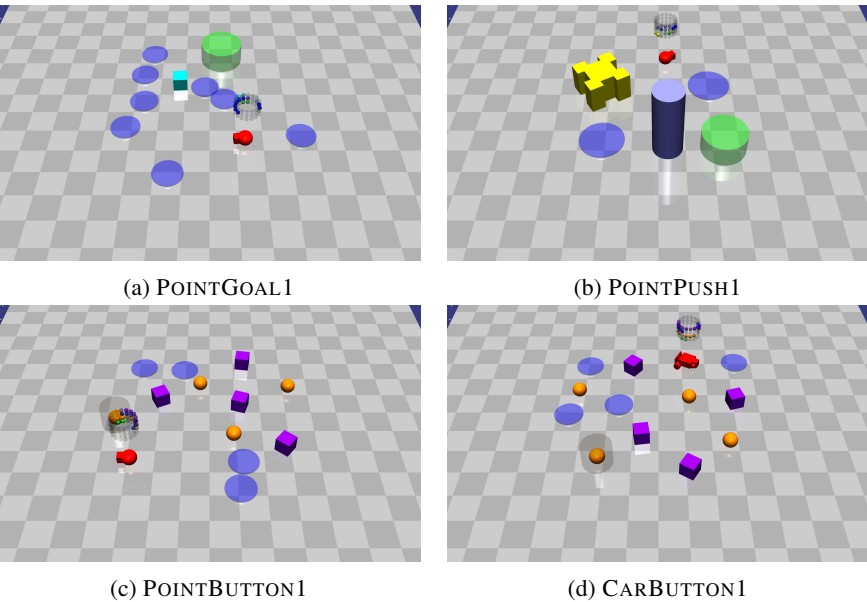

(a) POINTGOAL1        (b) POINTPUSH1

(c) POINTBUTTON1        (d) CARBUTTON1

Figure 7: Sample default task's from OpenAI's Safety Gym(nasium) environments (Ray et al., 2019; Ji et al., 2023). We use these to investigate the effect of termination in complex, high-dimensional, continuous control tasks. In all of the default tasks, $\mathcal{G} = \emptyset$ by default. (a) Here, a simple robot must navigate to a goal location 🟢 across a 2D plane while avoiding several hazards 🔵. The agent's sensors, actions, and rewards are identical to the PILLAR domain. Unlike the PILLAR domain, the goal location is randomly reset when the agent reaches it, but does not terminate the episode. (b) This task is similar to POINTGOAL1, but with the addition of a pillar obstacle 🔵 and a large box 🟨 the agent must push to the goal location 🟢 to receive the goal reward. (c-d) These tasks are also similar to POINTGOAL1, but with the more complex car robot for CARBUTTON1 and the addition of: (i) *Gremlins* 🟦, which are dynamic obstacles that move around the environment and must be avoided; and (ii) Buttons 🟢, where the agent must reach the goal button with a cylinder 🟢 to receive the goal reward.

# E  SAFETY-GYM PILLAR RESULTS WITH RAY ET AL. (2019) BASELINES

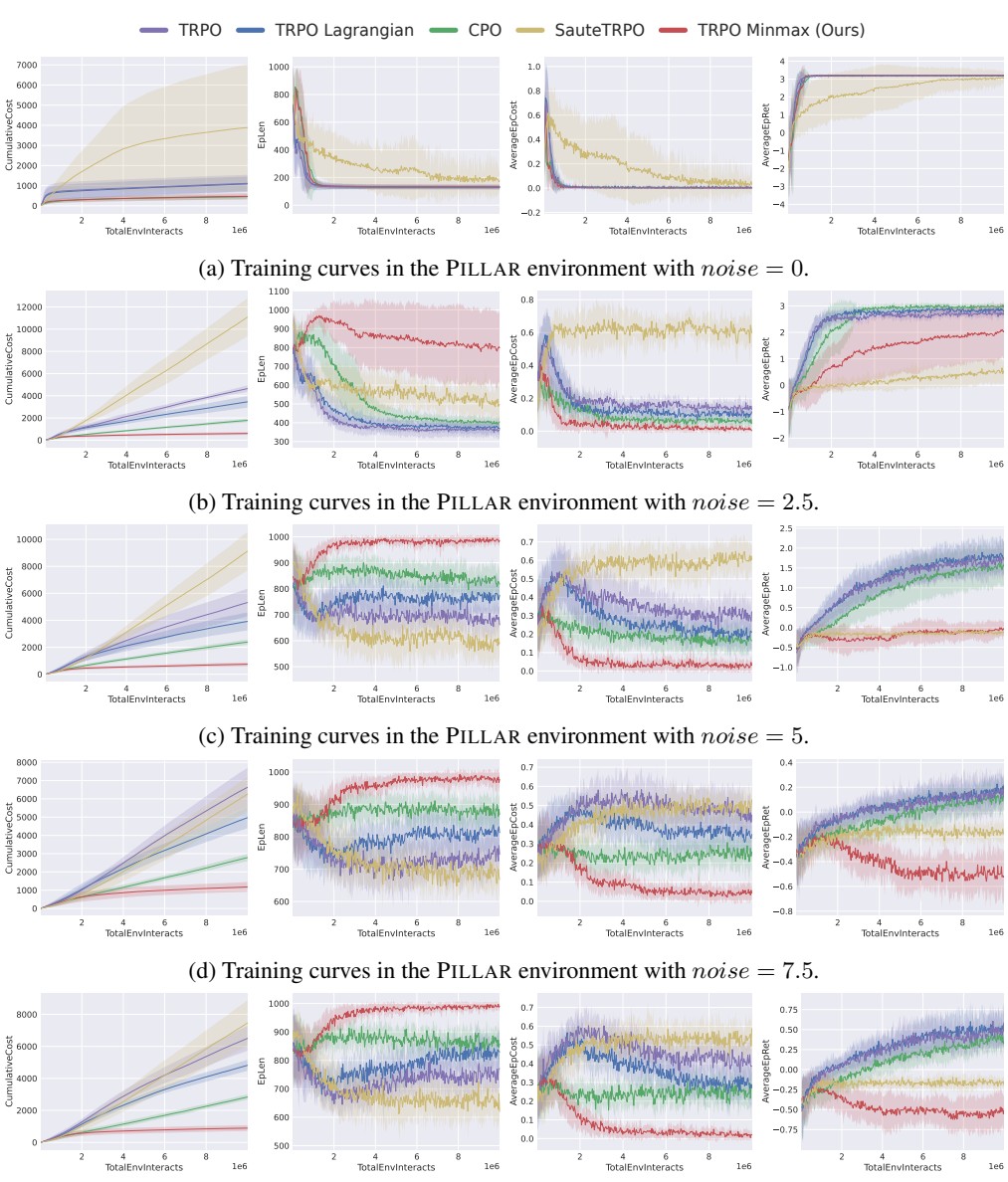

(a) Training curves in the PILLAR environment with $noise = 0$.

(b) Training curves in the PILLAR environment with $noise = 2.5$.

(c) Training curves in the PILLAR environment with $noise = 5$.

(d) Training curves in the PILLAR environment with $noise = 7.5$.

(e) Training curves in the PILLAR environment with $noise = 10$.

Figure 8: Training curves with Ray et al. (2019) baselines in the PILLAR environment. **(left)** Cummulative cost, **(middle-left)** Episode length, **(middle-right)** Failure rate, **(right)** Average returns. Since the environment is noisy, higher episode lengths are better (↑) because that means choosing safer longer paths (except for $noise = 0$). We observe that in the deterministic case $noise = 0$, all the algorithms achieve similar performance (except Sauté-TRPO), successfully maximising returns while minimising the failure rates. However, for the stochastic cases $noise > 0$, we can observe that all the baselines except Sauté-TRPO achieve significantly high returns at the expense of a rapidly increasing cumulative cost. Finally, by examining the episode length and failure rates for all the baselines in the stochastic cases, we can conclude that they have all learned risky policies that maximise rewards over short trajectories that are highly likely to result in collisions. In contrast, the results obtained show that TRPO-Minmax successfully solves the tasks while minimising cost for both deterministic and stochastic environments, when the noise levels are not too high ($noise \in [0, 2.5]$). In addition, we can observe from the episode lengths that TRPO-Minmax chooses the shortest path to the goal when there is no noise, but chooses longer paths as the noise increases.

| Noise | Algorithm | Costs ↓ | Success Rate ↑ | Returns ↑ | Total Steps ↓ |
|---|---|---|---|---|---|
| 0.0 | TRPO | $\mathbf{0.00 \pm 0.00}$ | $\mathbf{1.00 \pm 0.00}$ | $\mathbf{3.21 \pm 0.00}$ | $130.30 \pm 14.94$ |
| | TRPO-Lagrangian | $\mathbf{0.00 \pm 0.01}$ | $\mathbf{1.00 \pm 0.01}$ | $3.20 \pm 0.02$ | $132.16 \pm 14.43$ |
| | CPO | $\mathbf{0.00 \pm 0.00}$ | $\mathbf{1.00 \pm 0.00}$ | $\mathbf{3.21 \pm 0.01}$ | $\mathbf{128.06 \pm 14.40}$ |
| | Sauté-TRPO | $0.04 \pm 0.19$ | $0.95 \pm 0.21$ | $3.09 \pm 0.55$ | $176.51 \pm 117.93$ |
| | TRPO-Minmax | $\mathbf{0.00 \pm 0.00}$ | $\mathbf{1.00 \pm 0.00}$ | $\mathbf{3.21 \pm 0.01}$ | $131.53 \pm 15.15$ |
| | | | | | Total Steps ↑ |
| 2.5 | TRPO | $0.18 \pm 0.03$ | $0.82 \pm 0.03$ | $2.58 \pm 0.12$ | $351.33 \pm 40.17$ |
| | TRPO-Lagrangian | $0.13 \pm 0.03$ | $0.86 \pm 0.02$ | $2.73 \pm 0.09$ | $364.41 \pm 32.24$ |
| | CPO | $0.08 \pm 0.03$ | $\mathbf{0.92 \pm 0.03}$ | $\mathbf{2.91 \pm 0.10}$ | $393.36 \pm 29.50$ |
| | Sauté-TRPO | $0.62 \pm 0.49$ | $0.16 \pm 0.37$ | $0.59 \pm 1.27$ | $484.24 \pm 340.57$ |
| | TRPO-Minmax | $\mathbf{0.02 \pm 0.02}$ | $0.47 \pm 0.38$ | $2.00 \pm 1.02$ | $\mathbf{799.41 \pm 181.46}$ |
| 5.0 | TRPO | $0.32 \pm 0.07$ | $\mathbf{0.41 \pm 0.16}$ | $1.66 \pm 0.43$ | $665.62 \pm 38.34$ |
| | TRPO-Lagrangian | $0.20 \pm 0.07$ | $0.39 \pm 0.16$ | $\mathbf{1.78 \pm 0.47}$ | $760.66 \pm 43.54$ |
| | CPO | $0.18 \pm 0.04$ | $0.27 \pm 0.21$ | $1.53 \pm 0.54$ | $807.28 \pm 51.38$ |
| | Sauté-TRPO | $0.62 \pm 0.49$ | $0.01 \pm 0.07$ | $-0.09 \pm 0.54$ | $594.09 \pm 363.81$ |
| | TRPO-Minmax | $\mathbf{0.05 \pm 0.03}$ | $0.00 \pm 0.00$ | $-0.00 \pm 0.19$ | $\mathbf{975.59 \pm 17.81}$ |
| 7.5 | TRPO | $0.43 \pm 0.06$ | $\mathbf{0.02 \pm 0.03}$ | $0.45 \pm 0.21$ | $726.97 \pm 31.42$ |
| | TRPO-Lagrangian | $0.30 \pm 0.06$ | $0.01 \pm 0.01$ | $\mathbf{0.55 \pm 0.18}$ | $806.91 \pm 41.44$ |
| | CPO | $0.28 \pm 0.04$ | $0.00 \pm 0.01$ | $0.38 \pm 0.13$ | $830.78 \pm 25.03$ |
| | Sauté-TRPO | $0.54 \pm 0.50$ | $0.00 \pm 0.03$ | $-0.15 \pm 0.48$ | $650.94 \pm 364.90$ |
| | TRPO-Minmax | $\mathbf{0.02 \pm 0.02}$ | $0.00 \pm 0.00$ | $-0.46 \pm 0.20$ | $\mathbf{989.69 \pm 7.78}$ |
| 10.0 | TRPO | $0.46 \pm 0.08$ | $\mathbf{0.00 \pm 0.00}$ | $0.13 \pm 0.11$ | $725.03 \pm 49.64$ |
| | TRPO-Lagrangian | $0.36 \pm 0.09$ | $\mathbf{0.00 \pm 0.00}$ | $\mathbf{0.17 \pm 0.09}$ | $789.52 \pm 42.68$ |
| | CPO | $0.27 \pm 0.06$ | $\mathbf{0.00 \pm 0.00}$ | $0.10 \pm 0.10$ | $859.58 \pm 30.94$ |
| | Sauté-TRPO | $0.46 \pm 0.50$ | $\mathbf{0.00 \pm 0.00}$ | $-0.18 \pm 0.48$ | $701.60 \pm 355.32$ |
| | TRPO-Minmax | $\mathbf{0.07 \pm 0.05}$ | $\mathbf{0.00 \pm 0.00}$ | $-0.48 \pm 0.20$ | $\mathbf{960.96 \pm 28.39}$ |

Table 3: Evaluation of trained models with Ray et al. (2019) baselines in the PILLAR environment with varying noise levels. For each algorithm in each noise level, we train using 10 random seeds for 10 million steps and evaluate the learned policies over 100 random seeds, for a total of 1000 evaluation episodes. We report the mean and standard errors of various performance metrics, **bolding** the ones with the best mean. Figure 8 shows the training curves. Here, higher episode lengths are better for $noise > 0$ because that means the policy is taking longer safer paths. We observe that only TPRO-Minmax prioritises minimising the probability of unsafe transitions, consistently achieving the lowest cost while trading off the rewards. It achieves the same highest success rate as the baselines only in the deterministic case, since the pure maximisation of rewards here doesn't come at the cost of higher unsafe transitions. It also does not completely ignore the rewards when the noise is not too large ($noise = 2.5$). We can also observe from the training curves of $noise = 2.5$ (Figure 8b) that TPRO-Minmax has not converged in its rewards performance and is still increasing.

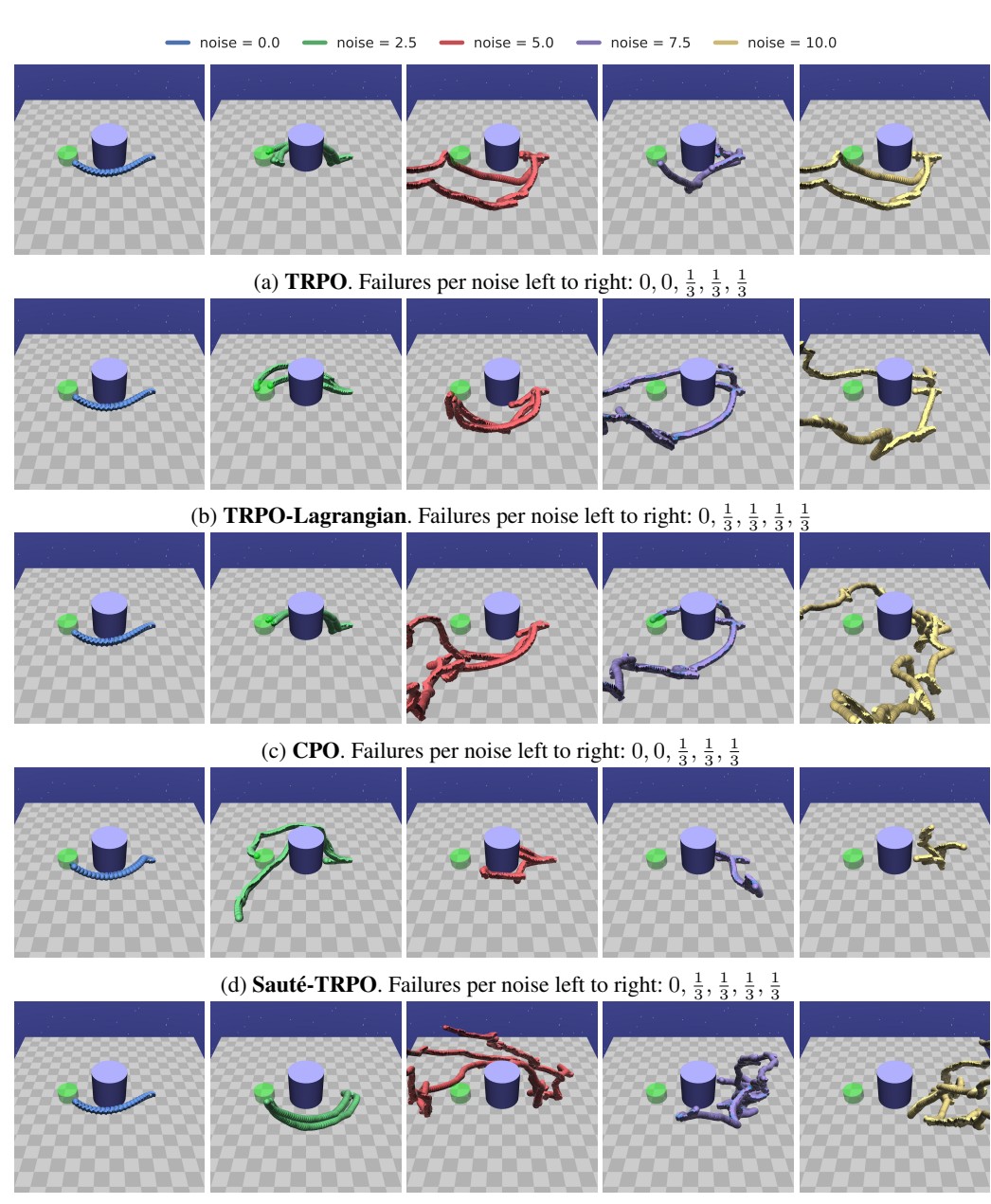

Figure 9: Sample trajectories of policies learned by each baseline and our **TRPO-Minmax** approach in the Safety Gym PILLAR environment with varying noise levels. To sample the trajectories for each noise level, we use the same three environment random seeds across all the algorithms. We can observe that $noise \geq 5$ is too noisy to learn safe policies, at least after 10 million training steps.

# F    ABLATIONS IN SAFETY-GYM WITH RAY ET AL. (2019) BASELINES

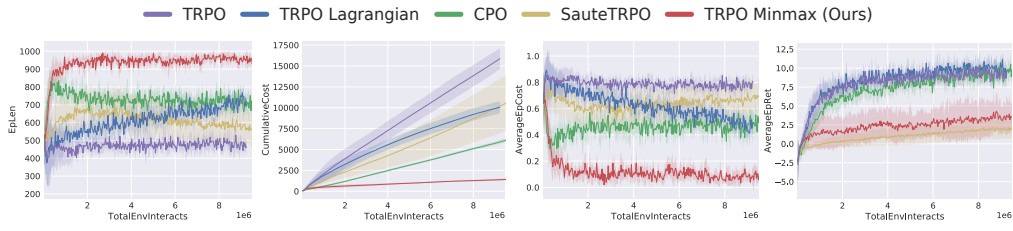

(a) Episode length (↑)    (b) Cumulative cost (↓)    (c) Failure rate (↓)    (d) Average returns (↑)

Figure 10: Comparison with baselines in POINTGOAL1, modified to terminate in $\mathcal{G} = \mathcal{G}^! = \{\bigcirc\}$. Here, higher episode lengths are better because episodes only terminate when the agent reaches $\mathcal{G}^!$ or after 1000 timesteps. Similar to Figure 8, all the baselines except Sauté-RL achieve significantly high returns at the expense of a rapidly increasing cumulative cost. By comparison, TRPO-Minmax dramatically reduces the failure rate while still being able to solve the task, as observed by average returns achieved as well as the trajectories observed. However, returns are lower since TRPO-Minmax learns safer longer paths to the goals (see sample trajectories in Figure 14).

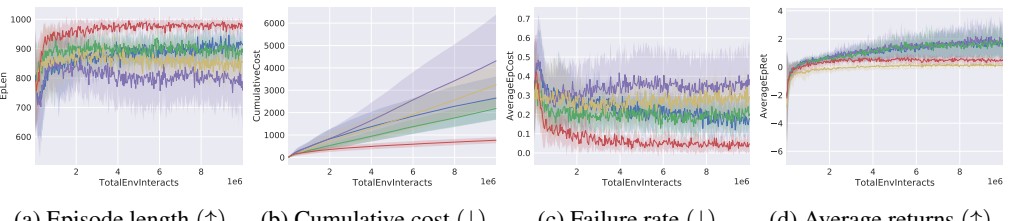

(a) Episode length (↑)    (b) Cumulative cost (↓)    (c) Failure rate (↓)    (d) Average returns (↑)

Figure 11: Comparison with baselines in POINTPUSH1, modified to terminate in $\mathcal{G} = \mathcal{G}^! = \{\bigcirc, \bigcirc\}$. Similar to Figure 8, the baselines achieve higher returns at the expense of a rapidly increasing cumulative cost while TRPO-Minmax consistently prioritises maintaining low failure rates by sacrificing rewards.

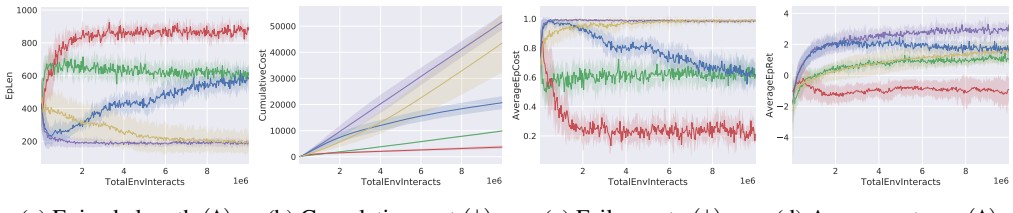

(a) Episode length (↑)    (b) Cumulative cost (↓)    (c) Failure rate (↓)    (d) Average returns (↑)

Figure 12: Comparison with baselines in POINTBUTTON1, modified to terminate in $\mathcal{G} = \mathcal{G}^! = \{\bigcirc, \blacksquare, \bullet\}$. Similar to Figure 8, the baselines achieve significantly high returns at the expense of a rapidly increasing cumulative cost while TRPO-Minmax consistently prioritises maintaining low failure rates.

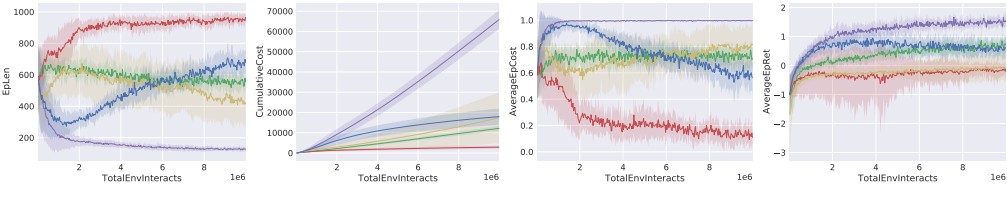

(a) Episode length (↑)    (b) Cumulative cost (↓)    (c) Failure rate (↓)    (d) Average returns (↑)

Figure 13: Comparison with baselines in CARBUTTON1, modified to terminate in $\mathcal{G} = \mathcal{G}^! = \{\bigcirc, \blacksquare, \bullet\}$). Similar to Figure 8, the baselines achieve significantly high returns at the expense of a rapidly increasing cumulative cost while TRPO-Minmax consistently prioritises maintaining low failure rates.

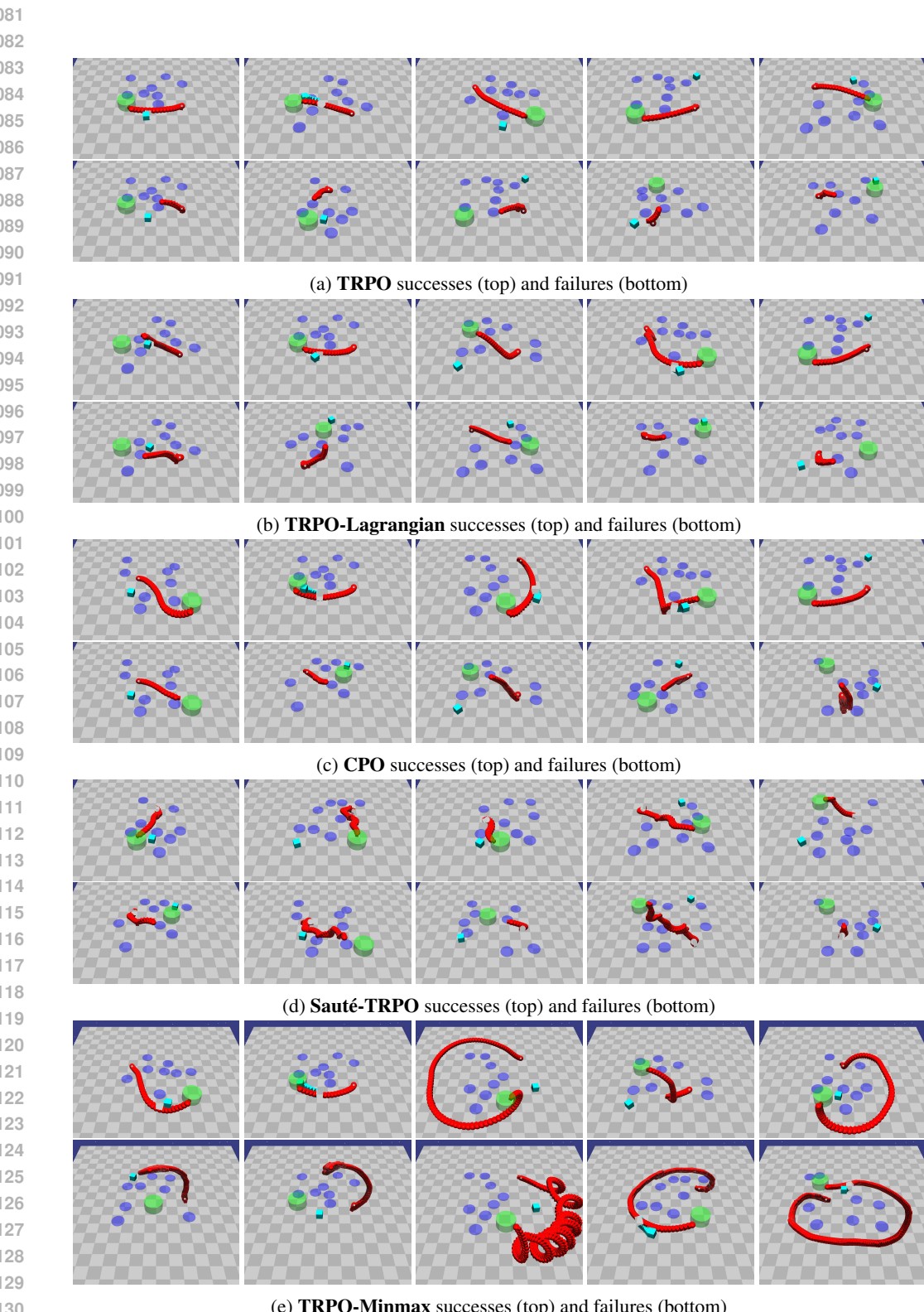

(a) **TRPO** successes (top) and failures (bottom)

(b) **TRPO-Lagrangian** successes (top) and failures (bottom)

(c) **CPO** successes (top) and failures (bottom)

(d) **Sauté-TRPO** successes (top) and failures (bottom)

(e) **TRPO-Minmax** successes (top) and failures (bottom)

Figure 14: Sample trajectories of policies learned by each baseline and our Minmax approach in the Safety Gym POINTGOAL1 domain, in the experiments of Figure 10. Trajectories that hit hazards or take more than 1000 timesteps to reach the goal location are considered failures.

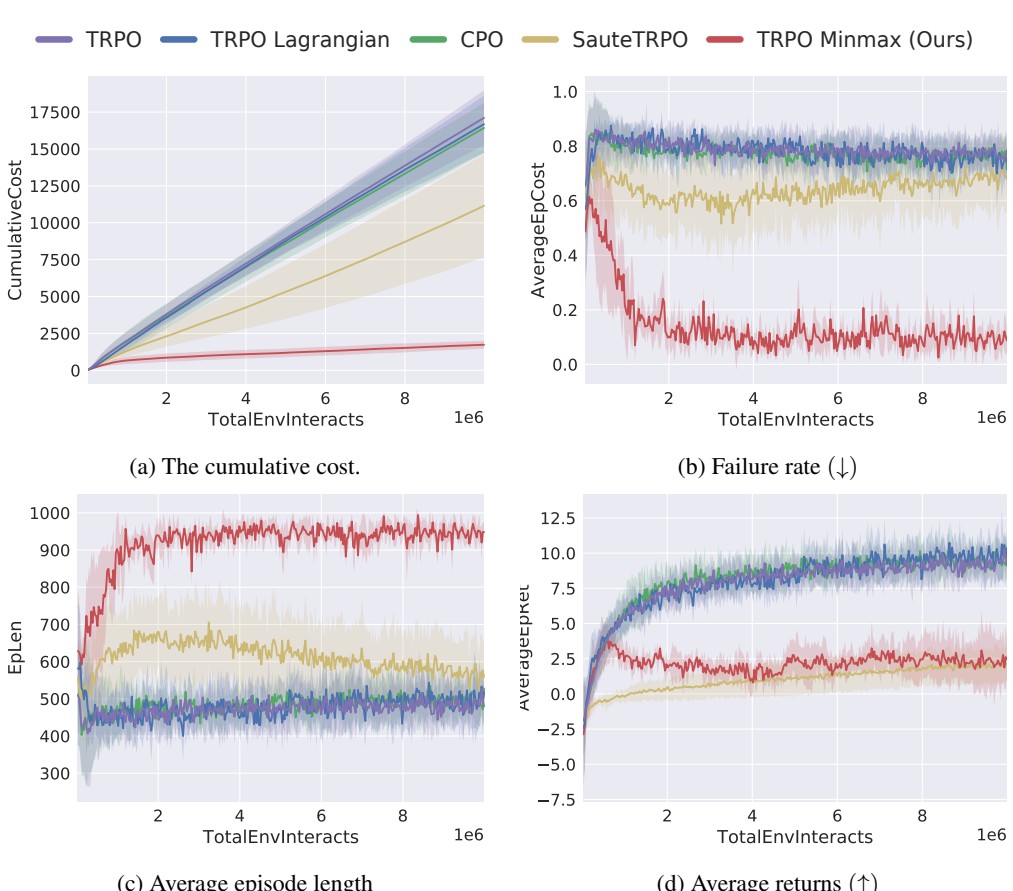

(a) The cumulative cost.

(b) Failure rate ($\downarrow$)

(c) Average episode length

(d) Average returns ($\uparrow$)

Figure 15: Comparison with baselines in POINTGOAL1, modified to terminate in $\mathcal{G} = \mathcal{G}^! = \{\bigcirc\}$. Here, higher episode lengths are better since episodes only terminate when the agent reaches a hazard or after 1000 timesteps. This experiment is similar to Figure 10, but instead of a cost threshold of 0, it uses a cost threshold of 25 for the baselines (as in Ray et al. (2019)) to check its effect on the performance of the baselines when episodes immediately terminate at unsafe states. We can observe drastically worse failure rates and cumulative costs for the baselines compared to their performance in Figure 10. Similar results where obtained when using a cost threshold of 1. These show how sensitive such approaches are to the cost threshold, while a reward only approach like TRPO-Minmax does not depend on such hyperparameters.

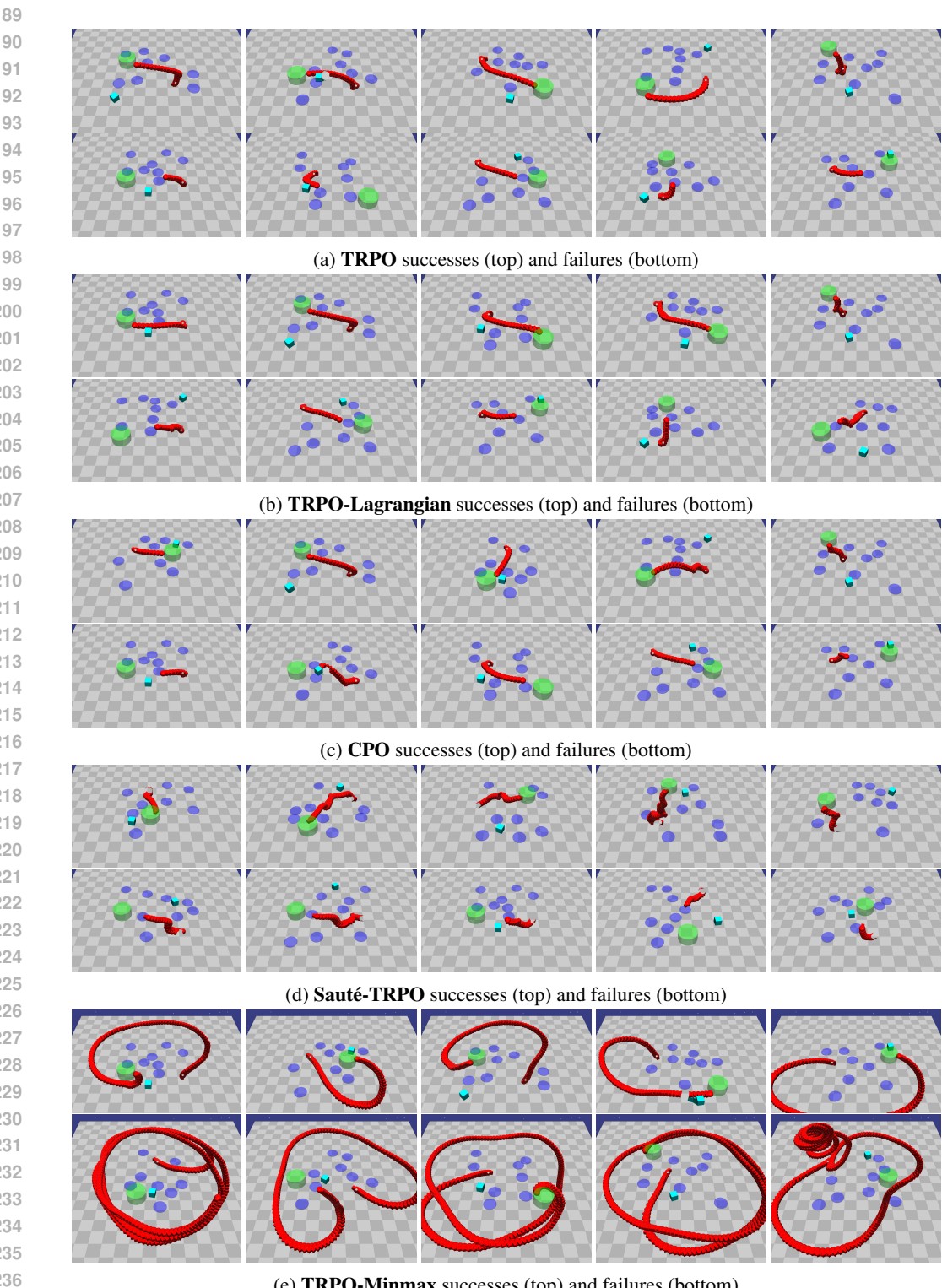

(a) **TRPO** successes (top) and failures (bottom)

(b) **TRPO-Lagrangian** successes (top) and failures (bottom)

(c) **CPO** successes (top) and failures (bottom)

(d) **Sauté-TRPO** successes (top) and failures (bottom)

(e) **TRPO-Minmax** successes (top) and failures (bottom)

Figure 16: Sample trajectories of policies learned by each baseline and our Minmax approach in the Safety Gym POINTGOAL1 domain, in the experiments of Figure 15. Trajectories that hit hazards or take more than 1000 timesteps to reach the goal location are considered failures.

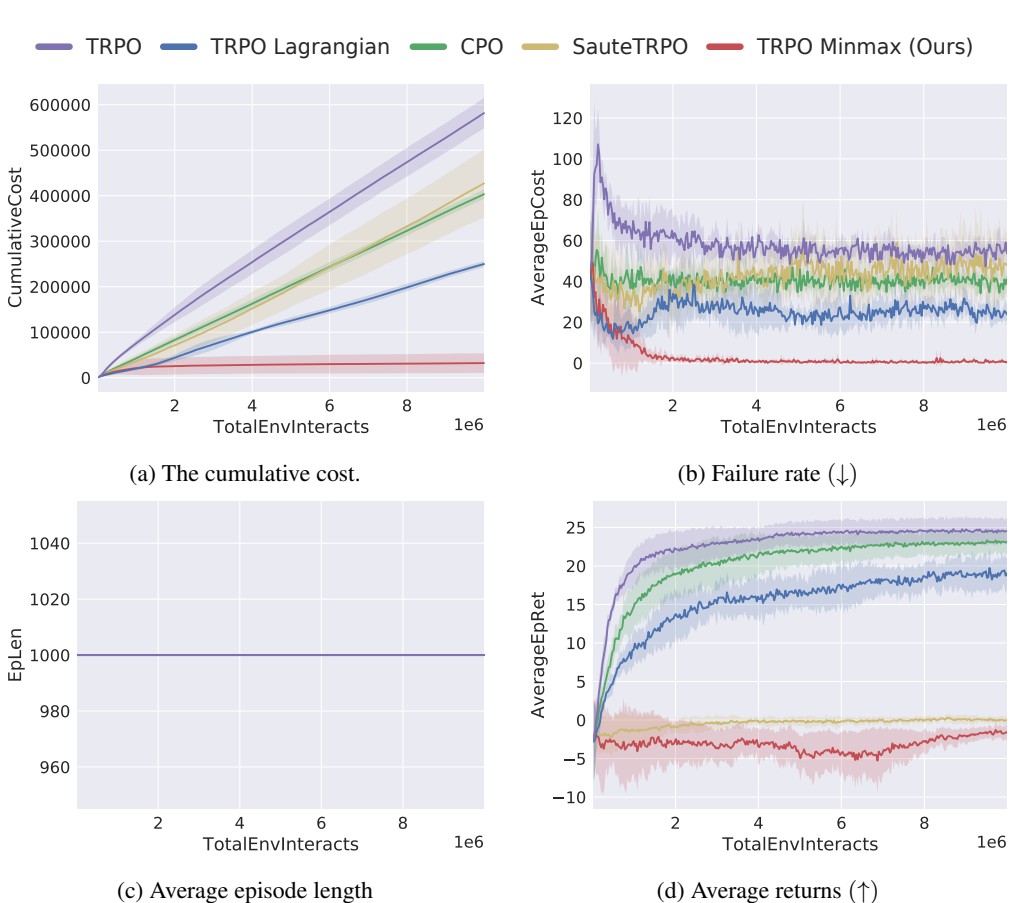

(a) The cumulative cost.

(b) Failure rate ($\downarrow$)

(c) Average episode length

(d) Average returns ($\uparrow$)

Figure 17: Comparison with baselines in the original Safety Gym POINTGOAL1 environment. Here, episodes do not terminate when a hazard is hit ($\mathcal{G} = \mathcal{G}^! = \emptyset$). Hence every episode only terminates after 1000 steps. We set the cost threshold for the baselines to 25 as in Ray et al. (2019). For TRPO-Minmax, we replace the reward with the Minmax penalty every time the agent is in an unsafe state (that is every time the cost is greater than zero), as in previous experiments and as per Algorithm 1. While TRPO-Minmax still beats the baselines in safe exploration (a-b), unlike the previous results with termination (Figure 15), it struggles to maximise rewards while avoiding unsafe states (d).

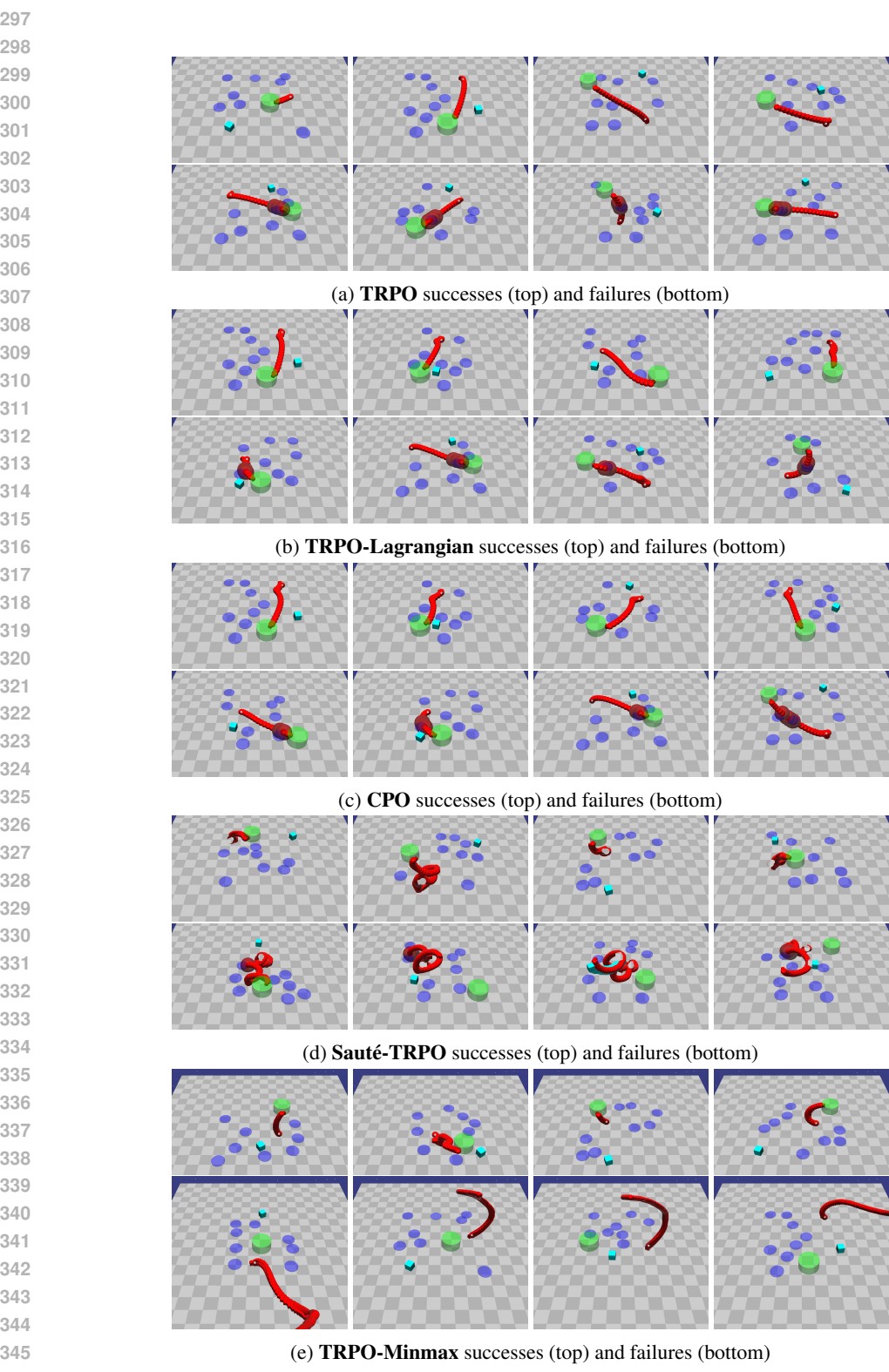

(a) **TRPO** successes (top) and failures (bottom)

(b) **TRPO-Lagrangian** successes (top) and failures (bottom)

(c) **CPO** successes (top) and failures (bottom)

(d) **Sauté-TRPO** successes (top) and failures (bottom)

(e) **TRPO-Minmax** successes (top) and failures (bottom)

Figure 18: Sample trajectories of policies learned by each baseline and our Minmax approach in the Safety Gym POINTGOAL1 domain, in the experiments of Figure 17. Trajectories that hit hazards (the hits are highlighted by the red spheres) or take more than 1000 timesteps to reach the goal location are considered failures.

# G Safety-Gym Pillar Training and Testing results with OmniSafe (Ji et al., 2024) baselines

| Noise | Algorithm | Costs ↓ | Success Rate ↑ | Returns ↑ | Total Steps ↓ |
|---|---|---|---|---|---|
| 0.0 | TRPO-Minmax (Ours) | **0.00 ± 0.00** | **1.00 ± 0.00** | **3.21 ± 0.00** | **136.60 ± 12.32** |
| | TRPO-Lagrangian | **0.00 ± 0.00** | **1.00 ± 0.00** | **3.21 ± 0.00** | 137.52 ± 14.50 |
| | Sauté-TRPO | **0.00 ± 0.00** | 0.99 ± 0.02 | 3.20 ± 0.03 | 142.88 ± 12.37 |
| | TRPO | **0.00 ± 0.00** | **1.00 ± 0.00** | **3.21 ± 0.00** | 138.92 ± 14.47 |
| | CPPOPID | **0.00 ± 0.00** | **1.00 ± 0.00** | **3.21 ± 0.00** | 140.70 ± 20.65 |
| | P3O | 0.10 ± 0.30 | 0.83 ± 0.35 | 2.74 ± 1.01 | 205.50 ± 220.31 |
| | | | | | **Total Steps ↑** |
| 1.5 | TRPO-Minmax (Ours) | **0.06 ± 0.02** | **0.94 ± 0.02** | **3.01 ± 0.08** | 262.19 ± 28.06 |
| | TRPO-Lagrangian | 0.09 ± 0.04 | 0.91 ± 0.04 | 2.90 ± 0.12 | 255.55 ± 26.62 |
| | Sauté-TRPO | 0.11 ± 0.04 | 0.89 ± 0.04 | 2.81 ± 0.14 | 232.26 ± 10.55 |
| | TRPO | 0.13 ± 0.08 | 0.87 ± 0.08 | 2.74 ± 0.29 | 262.91 ± 32.70 |
| | CPPOPID | 0.17 ± 0.08 | 0.83 ± 0.08 | 2.61 ± 0.31 | 344.60 ± 65.83 |
| | P3O | 0.11 ± 0.13 | 0.76 ± 0.33 | 2.43 ± 1.04 | **391.09 ± 221.08** |
| 2.5 | TRPO-Minmax (Ours) | **0.14 ± 0.05** | **0.80 ± 0.11** | **2.61 ± 0.27** | 503.49 ± 98.67 |
| | TRPO-Lagrangian | 0.20 ± 0.05 | 0.72 ± 0.24 | 2.38 ± 0.46 | 461.89 ± 132.78 |
| | Sauté-TRPO | 0.19 ± 0.09 | 0.76 ± 0.24 | 2.45 ± 0.54 | 435.18 ± 104.06 |
| | TRPO | 0.28 ± 0.10 | 0.63 ± 0.22 | 2.05 ± 0.52 | 446.21 ± 143.94 |
| | CPPOPID | 0.28 ± 0.16 | 0.66 ± 0.22 | 2.12 ± 0.64 | 485.50 ± 76.94 |
| | P3O | 0.17 ± 0.07 | 0.71 ± 0.16 | 2.42 ± 0.34 | **552.42 ± 139.28** |

Table 4: Evaluation of trained models with OmniSafe (Ji et al., 2024) baselines in the Pillar environment with varying noise levels. For valid comparison, TRPO-Minmax here is implemented by using Algorithm 1 with OmniSafe's implementation of TRPO. For each algorithm in each noise level, we train using 10 random seeds for 10 million steps and evaluate the learned policies over 100 random seeds, for a total of 1000 evaluation episodes. We report the mean and standard errors of various performance metrics, **bolding** the ones with the best mean. Figures 19-24 shows the training curves, including other noise levels for only TRPO-Minmax, TRPO-Lagrangian, and P3O. Here, higher episode lengths are better because that means the policy is taking longer safer paths. Given that, we observe that only TPRO-Minmax prioritises minimising the probability of unsafe transitions, consistently achieving the lowest cost while trading off the rewards. Interestingly, by using Algorithm 1 with OmniSafe's implementation of TRPO, TPRO-Minmax achieves the lowest cost, highest success rate, and highest returns across all noise levels.

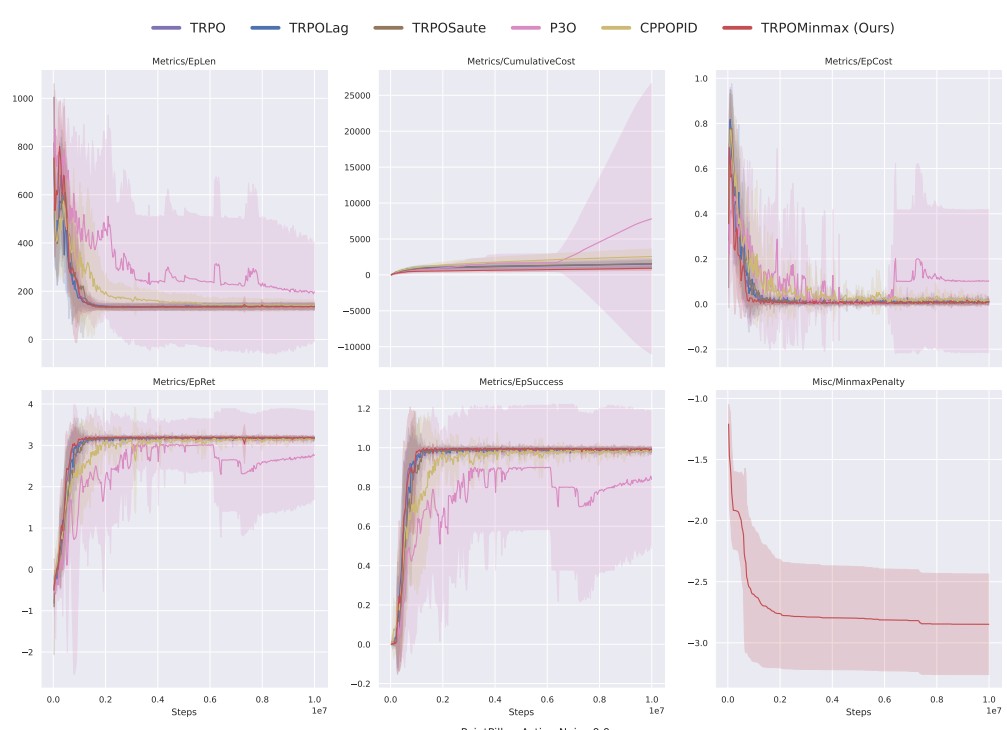

Figure 19: Training curves using OmniSafe in the PILLAR environment with $noise = 0$

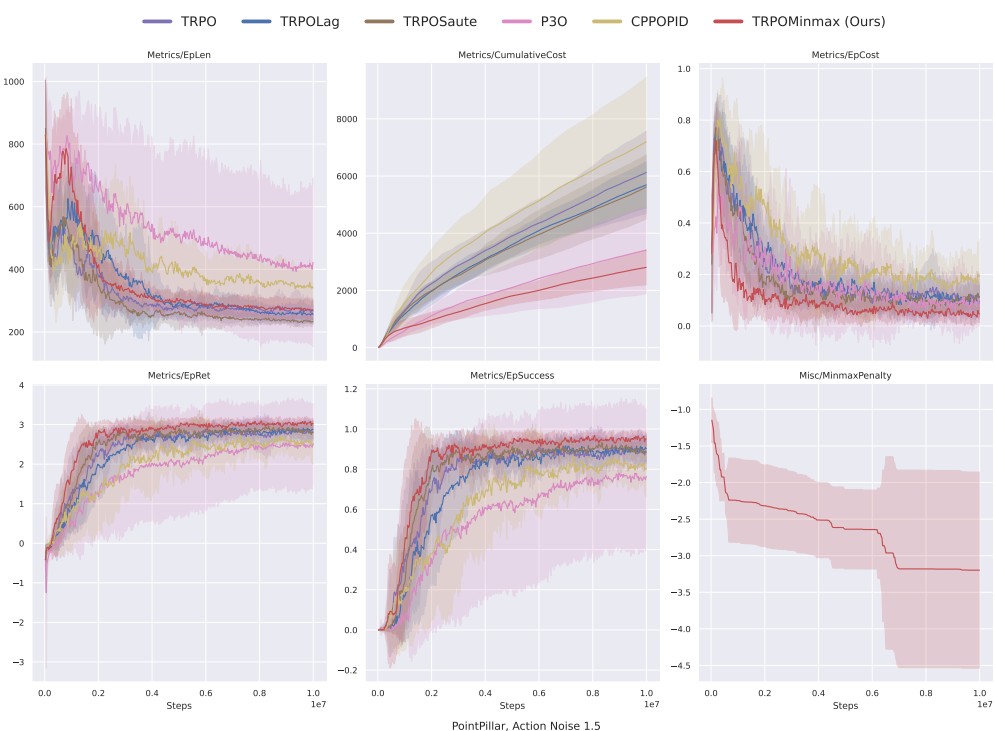

Figure 20: Training curves using OmniSafe in the PILLAR environment with $noise = 1.5$

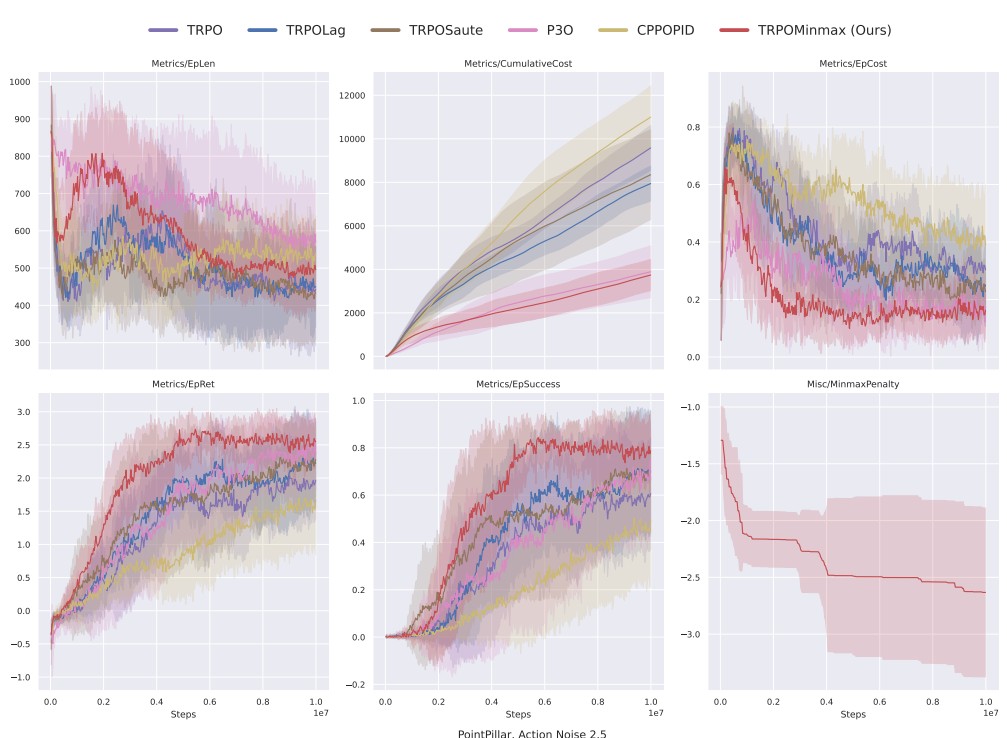

Figure 21: Training curves using OmniSafe in the PILLAR environment with $noise = 2.5$

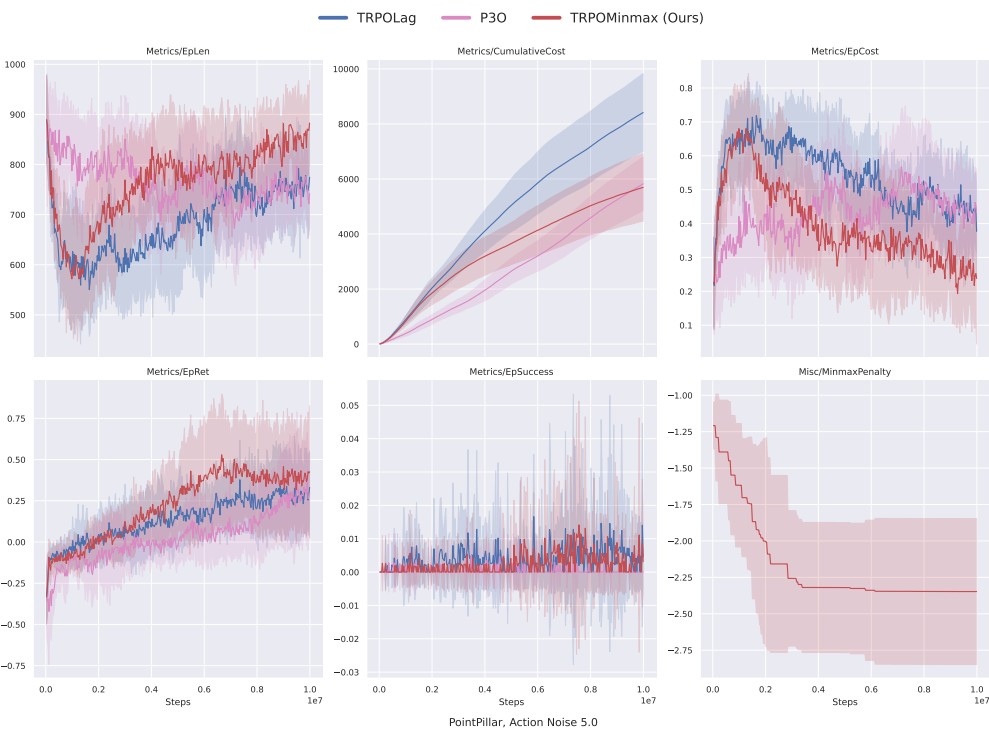

Figure 22: Training curves using OmniSafe in the PILLAR environment with $noise = 5$

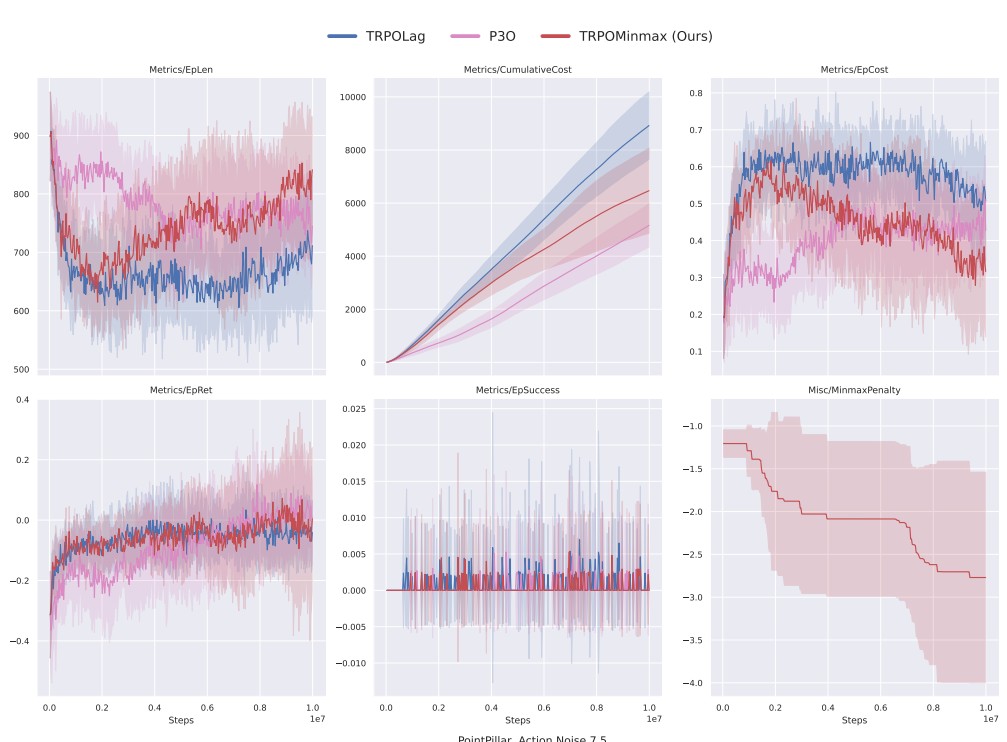

Figure 23: Training curves using OmniSafe in the PILLAR environment with $noise = 7.5$

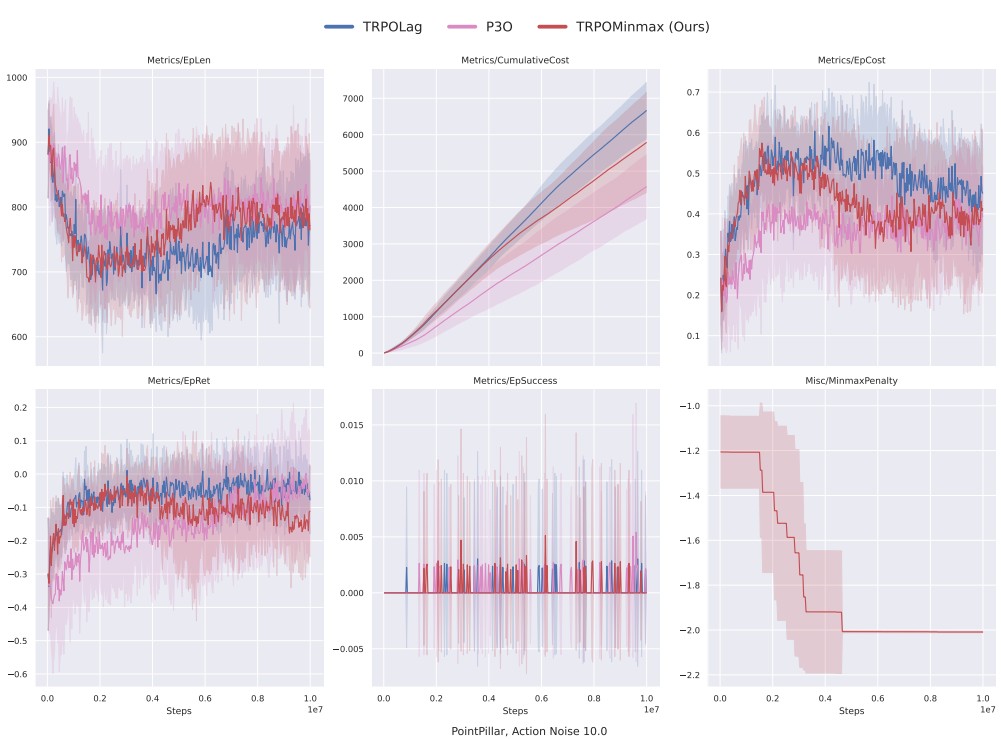

Figure 24: Training curves using OmniSafe in the PILLAR environment with $noise = 10$

# H  ABLATIONS IN SAFETY-GYMNASIUM DEFAULT ENVIRONMENTS WITH OMNISAFE (JI ET AL., 2024) BASELINES

| Algorithm | Costs ↓ | Success Rate ↑ | Returns ↑ | Total Steps |
|---|---|---|---|---|
| TRPO-Minmax (Ours) | **0.04 ± 0.03** | 0.50 ± 0.17 | 0.84 ± 0.45 | 532.08 ± 148.18 |
| PPO-Minmax (Ours) | **0.04 ± 0.02** | 0.84 ± 0.06 | 1.64 ± 0.18 | 253.69 ± 49.72 |
| TRPO-Lagrangian | 0.09 ± 0.03 | 0.86 ± 0.03 | 1.76 ± 0.08 | 119.18 ± 19.82 |
| Sauté-TRPO | 0.12 ± 0.03 | 0.87 ± 0.03 | 1.77 ± 0.09 | 77.97 ± 10.33 |
| TRPO | 0.10 ± 0.02 | 0.90 ± 0.02 | 1.84 ± 0.04 | 73.86 ± 4.37 |
| P3O | 0.08 ± 0.02 | **0.91 ± 0.02** | **1.86 ± 0.06** | 101.98 ± 13.03 |

Table 5: Evaluation of trained models with Ji et al. (2024) OmniSafe baselines in Safety-Gymnasium POINTGOAL1, modified to terminate in $\mathcal{G} = \{🟢, ⬤\}$ where $\mathcal{G}^! = \{⬤\}$. Episodes terminate when the agent reaches $\mathcal{G}$ or after 1000 timesteps, but due to the large number of hazards, shorter or longer timesteps are better depending on the random positions of hazards. Given that, we observe that our approach consistently achieves the lowest cost while trading off the rewards.

| Algorithm | Costs ↓ | Success Rate ↑ | Returns ↑ | Total Steps ↑ |
|---|---|---|---|---|
| TRPO-Minmax (Ours) | **0.08 ± 0.05** | 0.45 ± 0.16 | 1.87 ± 1.60 | **950.31 ± 31.45** |
| PPO-Minmax (Ours) | 0.13 ± 0.05 | **0.63 ± 0.13** | 4.45 ± 2.45 | 927.75 ± 28.89 |
| TRPO-Lagrangian | 0.62 ± 0.07 | 0.34 ± 0.06 | 10.17 ± 0.77 | 607.44 ± 56.15 |
| Sauté-TRPO | 0.79 ± 0.03 | 0.21 ± 0.03 | **11.01 ± 0.55** | 493.01 ± 24.51 |
| TRPO | 0.78 ± 0.05 | 0.22 ± 0.05 | 10.68 ± 0.74 | 483.42 ± 33.07 |
| P3O | 0.56 ± 0.07 | 0.42 ± 0.07 | 10.63 ± 0.74 | 667.08 ± 49.14 |

Table 6: Evaluation of trained models with Ji et al. (2024) OmniSafe baselines in Safety-Gymnasium POINTGOAL1, modified to terminate in $\mathcal{G} = \mathcal{G}^! = \{⬤\}$. Here, higher episode lengths are better because episodes terminate only when the agent reaches $\mathcal{G}^!$ or after 1000 timesteps. Similarly to Table 1, we exclude CPO from our analysis (denoted by a *) since its results are not consistent with those of Ray et al. (2019) and Achiam et al. (2017). Given that, we observe that despite the absence of terminal safe goals, our approach still prioritises minimising the probability of unsafe transitions, consistently achieving the lowest cost while trading off the rewards.

| Algorithm | Costs ↓ | Success Rate ↑ | Returns ↑ | Total Steps |
|---|---|---|---|---|
| TRPO-Minmax (Ours) | 4.11 ± 4.34 | 0.10 ± 0.04 | -2.21 ± 1.52 | 1000.00 ± 0.00 |
| PPO-Minmax (Ours) | **3.38 ± 3.08** | 0.13 ± 0.05 | -3.18 ± 2.71 | 1000.00 ± 0.00 |
| TRPO-Lagrangian | 18.18 ± 5.03 | **0.48 ± 0.05** | 9.24 ± 2.21 | 1000.00 ± 0.00 |
| Sauté-TRPO | 4.49 ± 3.12 | 0.17 ± 0.12 | 0.03 ± 0.63 | 1000.00 ± 0.00 |
| TRPO | 52.90 ± 3.27 | 0.07 ± 0.02 | **27.16 ± 0.07** | 1000.00 ± 0.00 |
| P3O | 30.72 ± 56.92 | 0.05 ± 0.03 | -1.18 ± 0.79 | 1000.00 ± 0.00 |

Table 7: Evaluation of trained models with Ji et al. (2024) OmniSafe baselines the Safety-Gymnasium POINTGOAL1, modified to terminate in $\mathcal{G} = \mathcal{G}^! = \emptyset$. Here, every episode terminates only after 1000 timesteps. Given that, we observe that despite no termination in the environment, our approach still achieves the lowest cost.

| Algorithm | Costs ↓ | Success Rate ↑ | Returns ↑ | Total Steps |
|---|---|---|---|---|
| TRPO-Minmax (Ours) | **0.08 ± 0.03** | 0.01 ± 0.01 | 0.47 ± 0.11 | 940.58 ± 20.33 |
| PPO-Minmax (Ours) | 0.12 ± 0.07 | **0.09 ± 0.14** | **1.12 ± 1.30** | 927.77 ± 31.18 |
| TRPO-Lagrangian | 0.12 ± 0.05 | 0.03 ± 0.03 | 0.62 ± 0.21 | 914.53 ± 29.49 |
| Sauté-TRPO | 0.13 ± 0.05 | 0.08 ± 0.14 | 0.92 ± 0.73 | 905.51 ± 33.87 |
| TRPO | 0.14 ± 0.06 | 0.05 ± 0.06 | 0.72 ± 0.37 | 903.23 ± 37.82 |
| P3O | 0.13 ± 0.04 | 0.06 ± 0.05 | 0.76 ± 0.33 | 921.17 ± 26.78 |

Table 8: Evaluation of trained models with Ji et al. (2024) OmniSafe baselines in Safety-Gymnasium POINTPUSH1, modified to terminate in $\mathcal{G} = \{$🟢,⚪,🔵$\}$ where $\mathcal{G}^! = \{$⚪,🔵$\}$. Episodes terminate when the agent reaches $\mathcal{G}$ or after 1000 timesteps, but due to the large object the agent needs to push to the goal while avoiding both hazards and the pillar, shorter or longer timesteps are better depending on the random positions of the hazards and pillar. Given that, we observe that our approach consistently achieves the lowest cost while obtaining the highest success rate and rewards.

| Algorithm | Costs ↓ | Success Rate ↑ | Returns ↑ | Total Steps ↑ |
|---|---|---|---|---|
| TRPO-Minmax (Ours) | **0.09 ± 0.03** | 0.05 ± 0.04 | 0.53 ± 0.15 | **905.26 ± 20.12** |
| PPO-Minmax (Ours) | 0.10 ± 0.02 | 0.03 ± 0.02 | 0.52 ± 0.07 | 914.64 ± 16.92 |
| TRPO-Lagrangian | 0.11 ± 0.03 | 0.13 ± 0.18 | 0.83 ± 0.49 | 844.21 ± 110.39 |
| Sauté-TRPO | 0.12 ± 0.05 | 0.10 ± 0.12 | 0.68 ± 0.30 | 838.98 ± 106.42 |
| TRPO | 0.15 ± 0.07 | **0.16 ± 0.21** | **0.86 ± 0.56** | 795.70 ± 157.18 |
| P3O | 0.13 ± 0.06 | 0.11 ± 0.12 | 0.78 ± 0.36 | 859.75 ± 74.76 |

Table 9: Evaluation of trained models with Ji et al. (2024) OmniSafe baselines in Safety-Gymnasium POINTPUSH1, modified to terminate in $\mathcal{G} = \mathcal{G}^! = \{$⚪,🔵$\}$. Here, higher episode lengths are better because episodes terminate only when the agent reaches $\mathcal{G}^!$ or after 1000 timesteps. Given that, we observe that despite the absence of terminal safe goals, our approach still prioritises minimising the probability of unsafe transitions, consistently achieving the lowest cost while trading off the rewards.

| Algorithm | Costs ↓ | Returns ↑ | Total Steps ↑ |
|---|---|---|---|
| TRPO-Minmax | **0.01 ± 0.01** | 610.50 ± 53.62 | 119.61 ± 10.06 |
| TRPO-Lagrangian | 0.09 ± 0.04 | 800.09 ± 100.72 | 147.62 ± 18.21 |
| Sauté-TRPO | 0.35 ± 0.12 | **957.11 ± 165.28** | **173.76 ± 28.66** |
| TRPO | 0.43 ± 0.18 | 944.04 ± 206.16 | 172.22 ± 35.41 |
| CPPOPID | 0.19 ± 0.06 | 805.04 ± 191.88 | 150.88 ± 33.30 |
| P3O | 0.02 ± 0.06 | 611.15 ± 42.52 | 121.90 ± 8.98 |

Table 10: Evaluation of Models for HUMANOIDVELOCITY (Figure 4d). The domain is only modified to terminate when an unsafe state is reached, which occurs when the velocity threshold is exceeded.

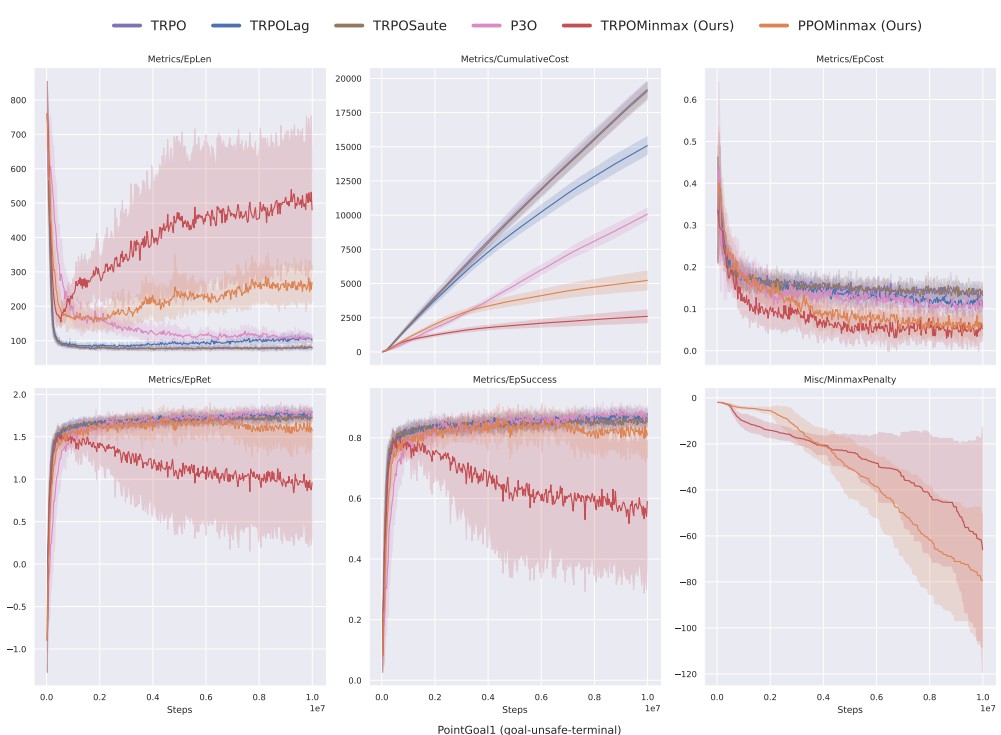

Figure 25: Training curves of models trained with Ji et al. (2024) OmniSafe baselines in the Safety-Gymnasium POINTGOAL1 environment, modified to terminate in $\mathcal{G} = \{\textcolor{green}{\bullet}, \textcolor{blue}{\bigcirc}\}$ where $\mathcal{G}^! = \{\textcolor{blue}{\bigcirc}\}$.

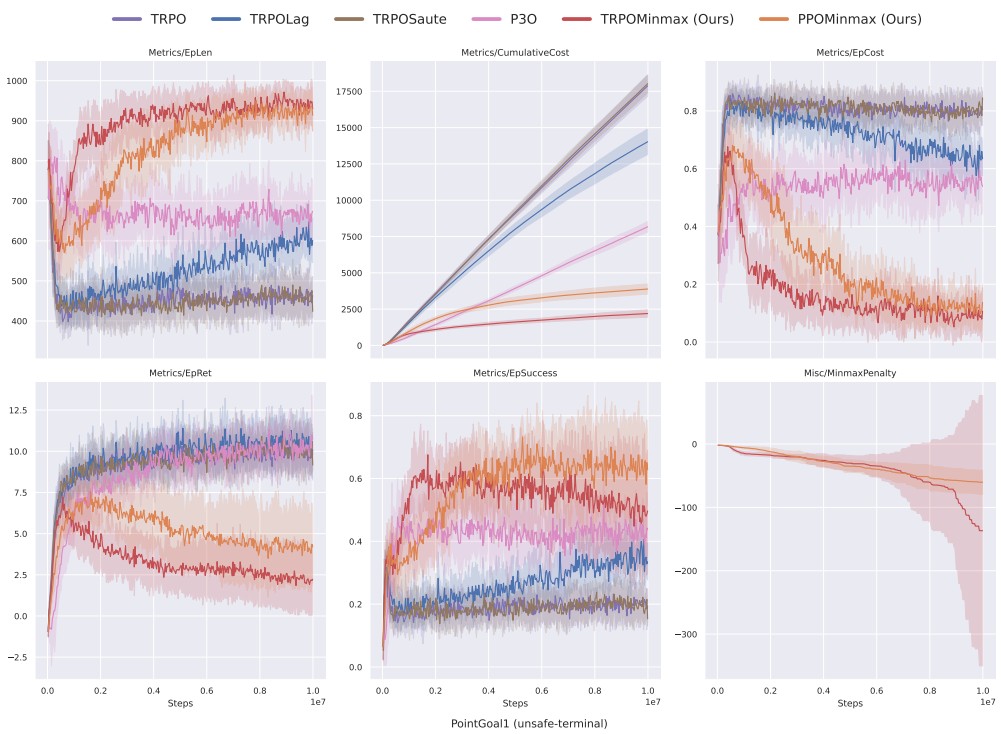

Figure 26: Training curves of models trained with Ji et al. (2024) OmniSafe baselines in the Safety-Gymnasium POINTGOAL1 environment, modified to terminate in $\mathcal{G} = \mathcal{G}^! = \{\textcolor{blue}{\bigcirc}\}$.

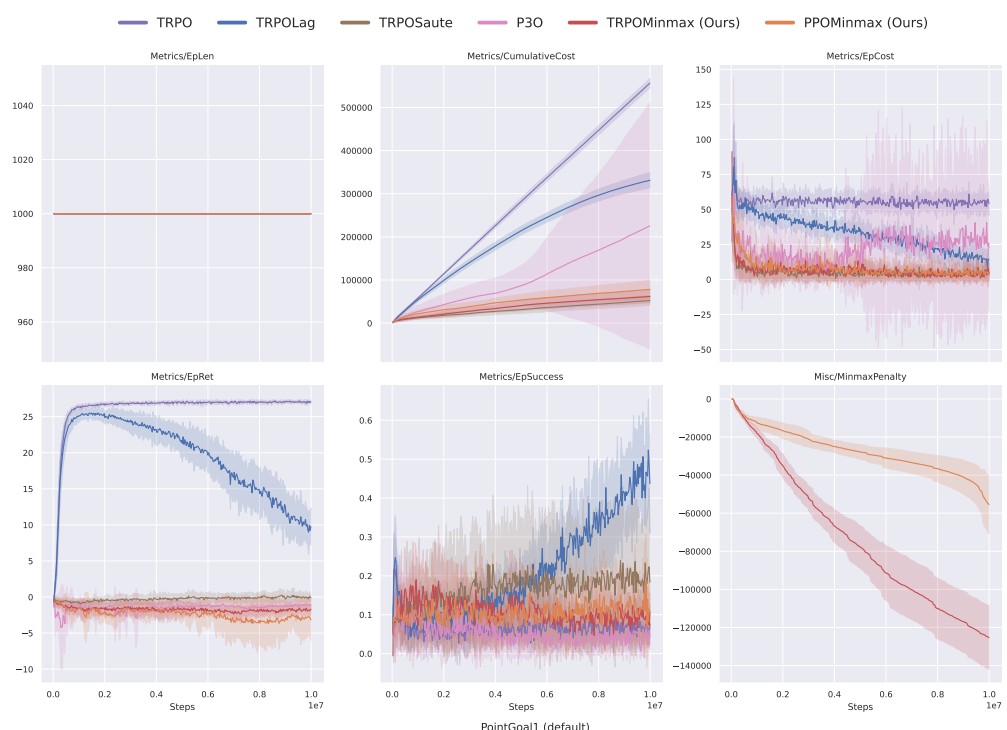

Figure 27: Training curves for trained models with Ji et al. (2024) OmniSafe baselines in the Safety-Gymnasium POINTGOAL1 environment, modified to terminate in $\mathcal{G} = \mathcal{G}^! = \emptyset$.

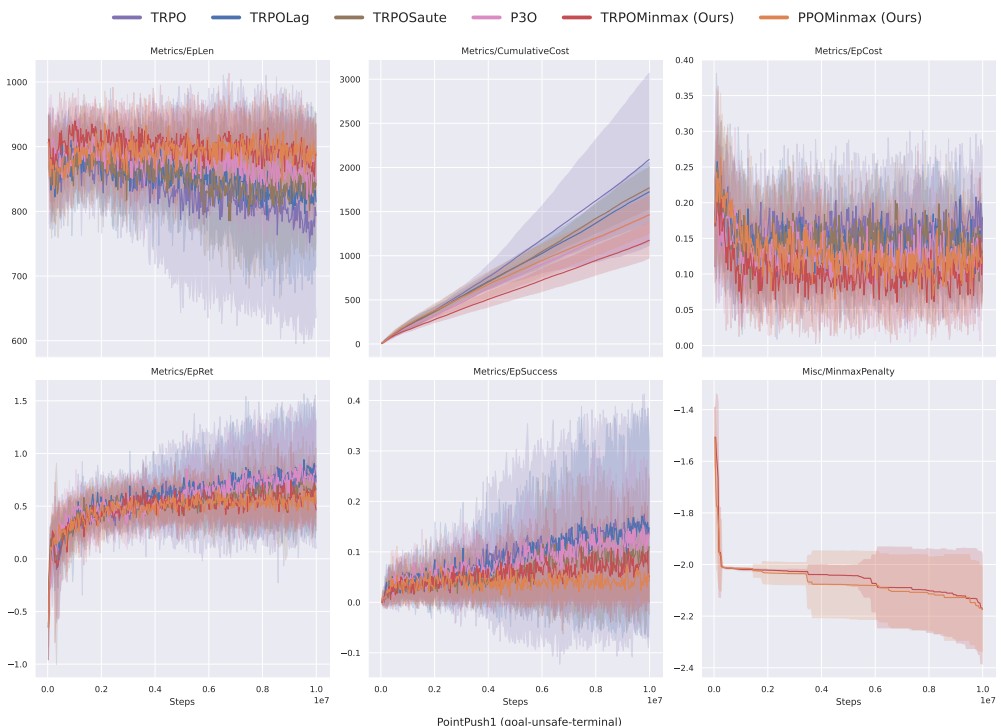

Figure 28: Training curves of models trained with Ji et al. (2024) OmniSafe baselines in the Safety-Gymnasium POINTPUSH1 environment, modified to terminate in $\mathcal{G} = \{\,\text{🟢},\text{⚪},\text{🔵}\,\}$ where $\mathcal{G}^! = \{\,\text{⚪},\text{🔵}\,\}$.

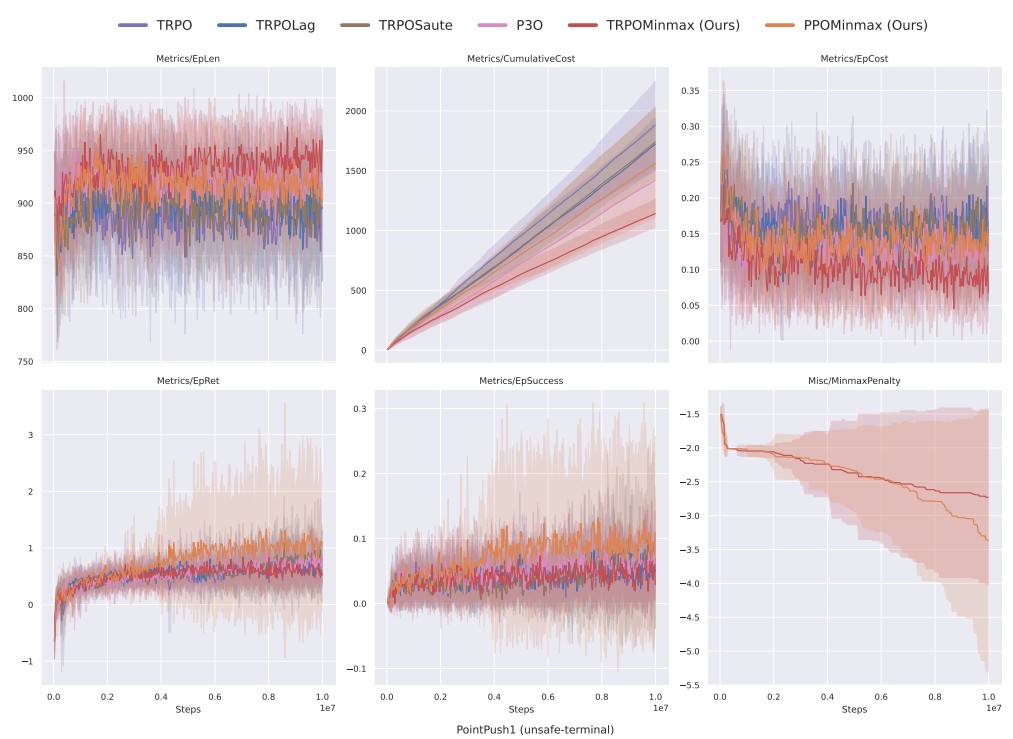

Figure 29: Training curves of models trained with Ji et al. (2024) OmniSafe baselines in the Safety-Gymnasium POINTPUSH1 environment, modified to terminate in $\mathcal{G} = \mathcal{G}^! = \{\,\bigcirc,\,\bigcirc\,\}$.

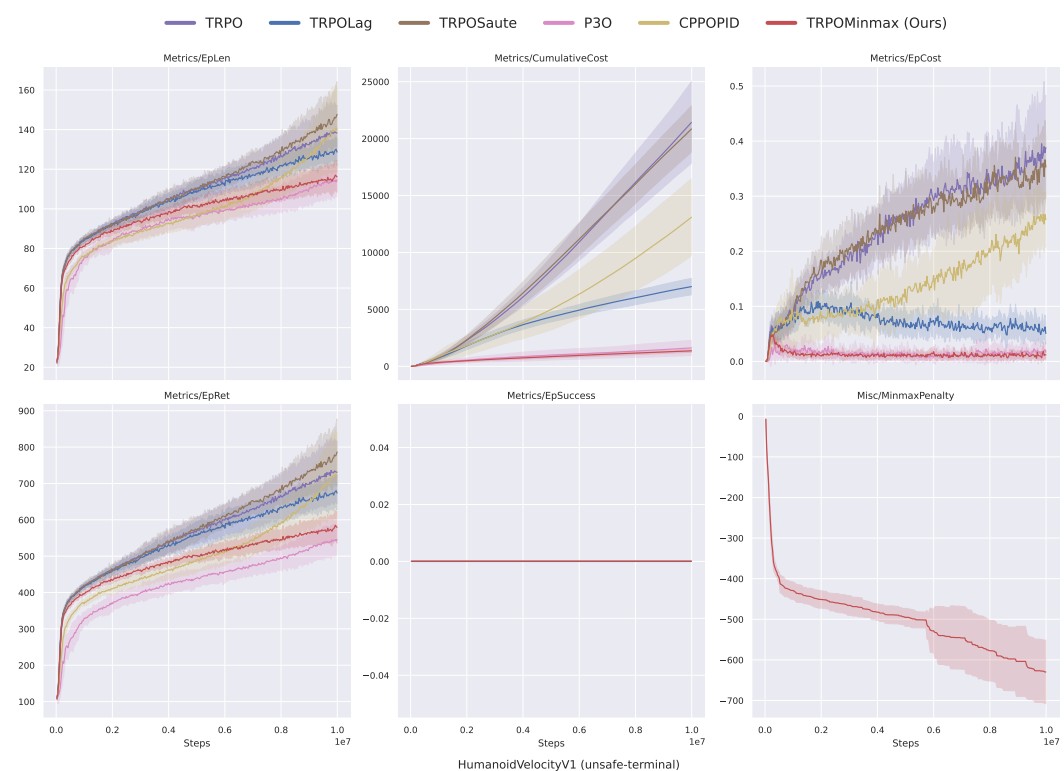

Figure 30: Training curves using OmniSafe in the HUMANOIDVELOCITY environment

