# OpenReview forum: "ROSARL: Reward-Only Safe Reinforcement Learning"
_ICLR.cc/2026/Conference — Submitted to ICLR 2026_

### Official Review · Reviewer_GG4v · 2025-10-16

**Soundness:** 3
**Presentation:** 3
**Contribution:** 3
**Rating:** 6
**Confidence:** 3

**Summary:**

This paper considers the problem of RL penalty reward design for safe reaching tasks. The setting is without discounted factors. The authors derive the calculation for the minimax penalty, the smallest penalty for unsafe states that leads to safe optimal policies. The authors first show the upper and lower bounds for this minimax penalty term based on task solvability and diameter, as well as the minimum and maximum rewards, then give a model-free practical estimate for the minimax penalty term using value estimates. The validity of their approach is first tested on the LAVA GRIDWORLD environment. Then they conduct experiments on the Safety Gym environment, and show that their proposed method can work better (less failure rate or cumulative cost with longer episode length) than baselines such as constrained RL, Lagrangian methods, and SauteTRPO.

**Strengths:**

1. The paper is well-written. The way the authors introduce these concepts is like a textbook, where abundant figures and examples provide a walkthrough for the key concepts.
2. The proposed method has strong theoretical guarantees.
3. Strong empirical results achieved compared to baselines. The experiments are conducted over 10-20 random seeds, showing statistical significance.

**Weaknesses:**

1. Missing baseline: the work does not compare with epigraph-based methods [1].
2. The simulation environment is just in 2D space (though observation state is 60D): not sure how the proposed method will behave in 3D space or for manipulation tasks.
3. Minor issue in writing: some citation issue at L193-194 and L316-317.


References:
1. So, Oswin, and Chuchu Fan. "Solving stabilize-avoid optimal control via epigraph form and deep reinforcement learning." arXiv preprint arXiv:2305.14154 (2023).

**Questions:**

The paper doesn't describe the limitations of the proposed method (beyond its applicability only to environments with unsafe terminal states). It will be great if the authors can comment on this.

---

> ### Author Response · Authors · 2025-12-03
>
> Thank you to the reviewer taking their time to review our paper. We have updated the paper with the requested additional results. We hope this and the following clarifications fully addresses the reviewer's concerns.
>
>  ## W1:
> > Missing baseline: the work does not compare with epigraph-based methods [1].
>
> Thanks you to the reviewer for the suggestion. We focused on comparing against the most popular baselines from prior works, as implemented in standard benchmark libraries like Safety Gym (Ray et al. 2019) and OmniSafe (Ji et al. 2024).
>
> However, since the reviewer's request is similar to the request from Reviewer yhcf, we have opted to add Reviewer yhcf suggested baseline (CPPOPID) since they said it is SOTA in constrained policy optimization. It is also already implemented for Gym(nasium)-based environments in the OmniSafe benchmark, while Epigraph does not have a publicly available implementation for Gym(nasium) environments. Our results (Table 1 page 9 of the main paper) show that similarly to the other baselines, CPPOPID successfully satisfies the constrains while minimising costs when there's no noise, but significantly struggles when there's noise. In general, we expect similar behaviours from other baselines like Epigraph which assume deterministic dynamics. We hope this addresses the reviewer's concern regarding additional baselines.
>
>
> ## W2
> > The simulation environment is just in 2D space (though observation state is 60D): not sure how the proposed method will behave in 3D space or for manipulation tasks.
>
> We have added the Humanoid safe velocity task (Figure 4.d page 7) in Table 2 of the main paper (with training curves in Figure 25 of the last page), which is representative of a 3D Safety-Gymnasium task as requested. Interestingly, even though this task is deterministic without noise, these results are still consistent with all our previous results showing that TRPO-Minmax is consistently the best at prioritising constraints satisfaction over reward maximisation, even better than the SOTA  suggested by Reviewer yhcf (see Table 10 page 31).
>
> ## W3
> > Minor issue in writing: some citation issue at L193-194 and L316-317.
>
> Thank you for the picking up these. We have fixed them in the updated paper.
>
> ## Q
> > The paper doesn't describe the limitations of the proposed method (beyond its applicability only to environments with unsafe terminal states). It will be great if the authors can comment on this.
>
> Beyond this limitation, another limitation mentioned in the paper (last paragraph of the main paper, page 9) is that our theoretical results are only derived for the stochastic shortest path setting. Interestingly, our practical algorithm still leads to strong constraints satisfaction in discounted settings, even outperforming SOTA baselines. This suggests that beyond the stochastic shortest path motivation for it, an interesting line of works will be a theoretical analysis of its behaviour in discounted settings and any other setting of interest (motivated by the empirical results obtained in this paper).

---

### Official Review · Reviewer_ci8A · 2025-10-26

**Soundness:** 2
**Presentation:** 2
**Contribution:** 2
**Rating:** 2
**Confidence:** 3

**Summary:**

This work presents a minmax penalty within TRPO algorithm, which applies the smallest penalty for unsafe states to generate safe policies.

**Strengths:**

The proposed security reinforcement learning algorithm has certain theoretical significance.

**Weaknesses:**

1. It has not been compared with existing reachability methods, which adopt the idea of minimax optimization.
2. The proposed method cannot guarantee absolute security of the strategy in theory or practice, which is crucial for secure reinforcement learning.
3. The comparison algorithm is relatively outdated.

**Questions:**

1. If Theorem 1 follows from the convergence guarantee of policy evaluation (Sutton & Barto, 1998), what is its significance?
2. What does Theorem 3 actually prove?
3. Can the proposed framework be combined with other reinforcement learning methods?

---

> ### Author Response · Authors · 2025-12-04
>
> We thank the reviewer for their time. Unfortunately, many of the reviewer's comments suggest fundamental misunderstandings of the paper, including key contributions, terminology, and even the problem setting. We clarify these issues below.
>
> # Reviewer Summary Mischaracterization and Missing Core Contributions
>
> The reviewer summarizes the paper as “presenting a minimax penalty within TRPO,” which omits the first two main contributions clearly stated in the introduction (page 2):
> * Introducing and theoretically bounding the Minmax penalty for safe RL, and
> * Proving that learning this quantity accurately is NP-hard, along with a constructive approach for estimation.
>
> The TRPO-based algorithm is only a demonstration of our third main contribution: A practical algorithm for learning the minmax penalty while learning a safe optimal policy. Hence TRPO is presented as a demonstration that our framework is algorithm-agnostic. The paper also shows this not just with TRPO, but also with Q-learning (main paper) and PPO (Appendix H page 30).
>
> Thus, the summary does not reflect the core content of the submission.
>
> # Incorrect References to “Security RL”
>
> The reviewer repeatedly refers to security reinforcement learning, a topic that is not mentioned anywhere in the paper.
> > "The proposed security reinforcement learning algorithm has certain theoretical significance."
>
> *Our work concerns safe RL, not security*. Because the terminology is central to the problem setting, these comments indicate a misunderstanding of the context and methodology described in the submission.
>
> # Claims Regarding Missing and Outdated Baselines
>
> The reviewer states that we “have not compared with existing reachability methods,” yet provides no examples or citations of the expected baselines. Our submission includes comparisons with standard baselines reported in popular benchmark suites (e.g., Ray et al. 2019, OmniSafe 2023): **CPO, TRPO, TRPO-Lagrangian, Sauté RL, and in the Appendix, P3O.**
>
> These represent the commonly accepted comparison set for safe RL algorithms. If the reviewer intended something different, it was not articulated. The reviewer also asks whether our framework can be combined with other RL algorithms. This is already demonstrated by experiments with Q-learning, TRPO, and PPO.
>
> # Incorrect Statement Regarding “No Guarantee of Safety”
>
> The reviewer asserts that the method “cannot guarantee absolute security,” again invoking terminology not used in the paper.
> If interpreted as safety, this contradicts the content of the submission:
> * Our first two contributions derive explicit optimality and safety bounds using the Minmax penalty and show how these can be accurately approximated using policy evaluation (Theorem 1, and Theorem 2).
>
> These results are central to the paper and appear to have been overlooked. The reviewer asks about the significance of Theorem 1 “since it follows from the convergence guarantee of policy evaluation.”, which suggests it was seen but may have been misread. This theorem’s purpose is not to restate known convergence results. Rather, it demonstrates that our safety bounds can be estimated directly via policy evaluation by choosing appropriate reward functions. This is important to obtain these precise bounds for a given MDP without needing closed-form derivations.
>
> The reviewer also asks what Theorem 3 proves. The text explains that Theorem 3 establishes NP-hardness of estimating the Minmax penalty, via a reduction to the longest path problem. Both questions suggest that the reviewer may have overlooked the explanations already provided in the paper.

---

### Official Review · Reviewer_PL6h · 2025-10-30

**Soundness:** 3
**Presentation:** 2
**Contribution:** 3
**Rating:** 6
**Confidence:** 4

**Summary:**

This paper proposes a new paradigm for Safe RL, whose core argument is that traditional CMDP methods, such as those using cost functions and Lagrangian multipliers, are unnecessary. Instead, the authors posit that the safety problem can be reformulated as a reward design problem.
The authors theoretically define a "Minmax penalty", which if assigned to all unsafe terminal states, ensures that the optimal policy of any standard, reward-maximizing RL algorithm will automatically be safe.
The authors derive theoretical upper and lower bounds for Minmax penalty, which depend on the "Diameter" (D) and "Solvability" (C). As this theoretical bound is difficult to compute in practice, the authors further propose a simple and model-free practical algorithm. This algorithm adaptively learns a sufficiently large penalty value by estimating the bounds of the value function online. Experiments demonstrate that combining this algorithm with standard RL algorithms achieves strong safety performance on benchmarks like Safety Gym, outperforming traditional constrained-optimization methods.

**Strengths:**

1. Reframing the episodic safety problem from a complex "constrained optimization" framework (CMDPs) back to a "reward design" problem is an insightful perspective.
2. The proposed practical algorithm is simple and easy to implement. It does not require manual tuning of hyperparameters. It can be used as a "plug-in" with any off-the-shelf, value-based RL algorithm, offering strong generality.
3. The method shows excellent performance in the Safety Gym experiments, particularly under high-noise settings.

**Weaknesses:**

1. The entire theoretical framework  is explicitly built on "undiscounted stochastic shortest path" (SSP) MDPs. However, the core experiments used to validate the algorithm  are conducted in "discounted," continuous-control, non-SSP environments. This makes the connection between the theoretical derivations and the experimental results weak.
2. The practical algorithm completely omits the solvability factor C from the theoretical bound. The authors claim the adaptive nature of the algorithm "implicitly" compensates for this, but this claim is not supported by any theory or ablation.
3. It is not at all clear how this method would generalize to the non-terminating CMDP setting. Figure 21 in the appendix seems to suggest the method's performance degrades in such a non-terminating setting.

**Questions:**

1. Given that the theory is for undiscounted SSPs, while the experiments are in discounted, continuous environments, can the authors provide deeper insight or a theoretical argument as to why Algorithm 1, which is derived from SSP theory, remains effective and robust in a discounted setting?
2.  Could the authors provide an ablation study to justify the omission of C? For example, in a simple tabular gridworld where C can be computed, how does Algorithm 1 (omitting C) compare to a policy trained using the "oracle" theoretical penalty that includes C?

---

> ### Author Response · Authors · 2025-12-03
>
> Thank you to the reviewer taking their time to review our paper. We have updated the paper with the requested additional results. We hope this and the following clarifications fully addresses the reviewer's concerns.
>
> ## W1,Q1
>
> > Given that the theory is for undiscounted SSPs, while the experiments are in discounted, continuous environments, can the authors provide deeper insight or a theoretical argument as to why Algorithm 1, which is derived from SSP theory, remains effective and robust in a discounted setting?
>
> The aim of our experiments was to evaluate the behaviour of our practical Algorithm in tasks which satisfy our theoretical setting (which we did in the Lava gridworld), and in tasks which which do not satisfy our theoretical setting (the Safety-Gym experiments). Note that as soon as we moved to function approximation and continuous control, our theoretical setting was already violated, regardless of whether it included discounting or not. Hence we focused on discounted settings since that is the setting assumed in most prior works.
>
> **Perhaps the best insight here comes from the theoretical similarities between the SSP setting and the discounted setting. As shown by Bertsekas (1987), every discounted setting can be converted into a stochastic shortest path one.** This is mainly because one can think of the discounting as essentially setting a horizon limit to the agent's expected rewards, which can then be thought of as an absorbing terminal state. We believe this also explains our ablation results over terminal states (Tables 5-7), were we see that the agent's ability to minimise the probability of unsafe transitions (i.e. when the cost>0) diminishes as the terminal states become less explicit (relies on the discounted horizon's terminal state). Despite this, we observe that our agent still outperforms all baselines in terms of number of unsafe transitions.
>
> ## W2,Q2
>
> > For example, in a simple tabular gridworld where C can be computed, how does Algorithm 1 (omitting C) compare to a policy trained using the "oracle" theoretical penalty that includes C?
>
> **As requested, we have added Q-learning using the  "oracle" theoretical penalty that includes C to our Lava gridworld experiments (Figure 5 page 7).** We observe that Algorithm matches the performance of the oracle in both failure rates and returns when there's no noise ($sp=0$). Interestingly, we also see that the oracle struggles to maximise rewards when there's noise ($sp=2.5$ and $sp=5$) due to the large theoretical penalties (while still successfully minimising faillure rates), possibly requiring much longer training time to converge. In contrast, Algorithm 1 still successfully learns to maximise rewards while minimise unsafe transitions, converging to the best performance at a similar rate as the deterministic case.
>
> # W3
>
> > It is not at all clear how this method would generalize to the non-terminating CMDP setting. Figure 21 in the appendix seems to suggest the method's performance degrades in such a non-terminating setting.
>
> Please note that Figure 21 (now Figure 17 page 24 in the updated paper) uses a cost threshold of 25 as mentioned in the caption (the agent is allowed reach unsafe states at 25 times within an episode). Hence it is simply there to replicate prior works for sanity check, showing the setting from prior works where we do not care about minimising the probability of reaching unsafe states (where cost>0). **Notice how all baselines drastically fail to achieve costs near zero, instead achieving costs greater than 20 (repeatedly entering unsafe states to maximise rewards)**
>
> For the non-terminating CMDP setting where we still care about achieving the lowest cost possible (cost threshold 0), please see Table 7 page 30 (and training curves in Figure 27 page 33). **Our results show that we are still the best at minimising costs, while even getting higher success rates an TRPO and P3O.**

---

### Official Review · Reviewer_yhcf · 2025-11-01

**Soundness:** 2
**Presentation:** 3
**Contribution:** 2
**Rating:** 4
**Confidence:** 3

**Summary:**

The paper presents an alternative to traditional constrained policy optimization methods by learning a penalty term for states that violate constraints, resulting in the learning of safe optimal policies. The paper derives its penalty term from the concepts of diameter and solvability, which are explained in the paper. The derived penalty requires knowledge of the environment's dynamics; thus, the paper provides a practical algorithm that sidesteps this issue and presents some experiments demonstrating the performance of their algorithm. The paper also provides an analysis of performance in cases where the assumptions hold and in other cases where they do not.

**Strengths:**

The paper's strengths lie in providing an alternative to unstable policy optimization methods by introducing a penalty term that eliminates the need for such approaches. The paper is well-presented and generally sound, albeit with some flaws. The analysis in a lower-dimensional environment, as well as the comparison between the performance of the practical algorithm and the environment where the method's assumption holds, is quite helpful.

**Weaknesses:**

The paper derives its penalty term using the concepts of diameter and solvability, which require knowledge of the dynamics. In the practical implementation of their method, which does not require knowledge of the dynamics. The empirical experiments are on the weaker side. The method underperforms Lagrangian TRPO in task performance. Also, the paper compares their method only with a single threshold; further, the method does not compare their approach with the PID Lagrangian method, which is SOTA in constrained policy optimization. Further, there's a serious flaw in how the paper motivates its approach; the authors provide reasoning that their approach offers an alternative to shaped constraint costs. However, in many constrained RL problems, the cost is considered to be sparse. In my view, this approach provides an alternative to the issue of constrained policy optimization, which can be unstable. This is a significant distinction.

Minor errors:
- "minimized when reaching"  in the abstract
- references in lines 193 and 215, 316

Stooke, Adam, Joshua Achiam, and Pieter Abbeel. "Responsive safety in reinforcement learning by pid lagrangian methods." International Conference on Machine Learning. PMLR, 2020.

**Questions:**

- Can you compare your approach to Lagrangian TPO under different constraint thresholds?
- Can you compare your algorithm to PID Lagrangian approaches?
- Can you run experiments on other safety gym domains?

I would be willing to raise my score if the authors can provide answers to my questions.

Stooke, Adam, Joshua Achiam, and Pieter Abbeel. "Responsive safety in reinforcement learning by pid lagrangian methods." International Conference on Machine Learning. PMLR, 2020.

---

> ### Author Response · Authors · 2025-12-03
>
> Thank you to the reviewer taking their time to review our paper. We have updated the paper with the requested additional results. We hope this and the following clarifications fully addresses the reviewer's concerns.
>
>  ## Q1:
> > Can you compare your approach to Lagrangian TPO under different constraint thresholds?
>
> Due to space constraints, we included this result in the appendix of the original submission (now Figure 15 page 22 in the updated paper). We observe drastically worse failure rates and cumulative costs for the baselines when using cost thresholds of 1 and 25, compared to their performance when using a threshold of 0. These show how sensitive such approaches are to the cost threshold when the goal maximal safety (i.e. to minimise the probability of unsafe transitions), while a reward only approach like TRPO-Minmax does not depend on such hyperparameters.
>
> ## Q2
> > Can you compare your algorithm to PID Lagrangian approaches?
>
> We have added the requested CPPOPID to our baselines, using the implementation from the OmniSafe benchmark. Our results show that similarly to the other baselines, CPPOPID successfully satisfies the constrains while minimising costs when there's no noise, but significantly struggles when there's noise.
>
> ## Q2
> > Can you run experiments on other safety gym domains?
>
> Due to space constraints, we included these results (using Ray et al.'s baselines implementation) in the appendix of the original submission (now Figures 10-13 page 20 in the updated paper). These results are consistent with our results in the Pillar domain using the same baselines.

---

### Meta-Review · Area_Chair_nCsG · 2026-01-07

**Summary:**

This paper proposes a new paradigm for Safe RL  by investigating the concept of a minmax penalty. Experiments demonstrate that the proposed approach algorithm outperforms traditional constrained-optimization methods on some benchmarks.
The major concerns raised by the reviewers include: (1) inconsistency between the derivation of the penalty term and the practical method implementation; (2) unconvincing experimental results without SOTA baselines; and (3) flaws in the paper’s motivation. The authors’ rebuttal provided additional, albeit partial, results with extra new baselines and experiments in 3D space, but failed to adequately address or clarify the other concerns (e.g., (1) and (3)).
Overall, the paper received relatively low reviewer scores and unpromising evaluations, which do not support acceptance.

**Reviewer Concerns:**

The authors’ rebuttal provided additional, albeit partial, results with extra new baselines and experiments in 3D space, but failed to adequately address or clarify the other concerns (e.g., (1) and (3)).

**Reviewer Scores:**

The rebuttal lacks sufficient clarity. I do not think there would be significant score changes.

---

### Decision · Program_Chairs · 2026-01-26

Reject